# Federated Oriented Learning: A Practical One-Shot Personalized Federated Learning Framework

**Guan Huang** [1]   **Tao Shu** [1]

## Abstract

Personalized Federated Learning (PFL) has become a promising learning paradigm, enabling the training of high-quality personalized models through multiple communication rounds between clients and a central server. However, directly applying traditional PFL in real-world environments where communication is expensive, limited, or infeasible is challenging, as seen in Low Earth Orbit (LEO) satellite constellations, which face severe communication constraints due to their high mobility, limited contact windows. To address these issues, we introduce Federated Oriented Learning (FOL), a novel four-stage one-shot PFL algorithm designed to enhance local model performance by leveraging neighboring models within stringent communication constraints. FOL comprises model pretraining, model collection, model alignment (via fine-tuning, pruning, post fine-tuning, and ensemble refinement), and knowledge distillation stages. We establish two theoretical guarantees on empirical risk discrepancy between student and teacher models and the convergence of the distillation process. Extensive experiments on datasets Wildfire, Hurricane, CIFAR-10, CIFAR-100, and SVHN demonstrate that FOL consistently outperforms state-of-the-art one-shot Federated Learning (OFL) methods; for example, it achieves accuracy improvements of up to 39.24% over the baselines on the Wildfire dataset.

## 1. Introduction

Recently, Personalized Federated Learning (PFL) has been proposed as an effective means to address the sub-optimal performance of the global model produced by federated learning (FL) (Li et al., 2020b; Yang et al., 2019; Huang

[1]Department of CSSE, Auburn University, Auburn, AL, 36849, USA. Correspondence to: Tao Shu <tshu@auburn.edu>.

*Proceedings of the 42$^{nd}$ International Conference on Machine Learning*, Vancouver, Canada. PMLR 267, 2025. Copyright 2025 by the author(s).

et al., 2024) when the model is used at local clients with heterogeneous (i.e., non-IID) datasets (Huang et al., 2023; Ye et al., 2023). Benefiting from the shared knowledge of the global model, PFL adapts it to form personalized local models at each client, which perform better than both the global model and those local models trained independently on each client's data. (Tan et al., 2022a; Deng et al., 2020; Collins et al., 2021). Existing works in PFL can be broadly categorized into two groups: (1) Parameter decoupling approaches, which divide the model into shared and personalized components. The shared backbone (feature extractor) captures generalizable knowledge across clients, while the personalized head (classifier) is specifically adapted to each client's unique data distribution (Liang et al., 2020; Oh et al., 2021; Tan et al., 2022b; Su et al., 2023a; Wu et al., 2024). (2) Optimization-based approaches, which employ advanced optimization techniques, such as regularization-based methods, dynamic aggregation, and second-order optimization, to constrain local models and adjust aggregation weights, thereby balancing global collaboration with local personalization (T Dinh et al., 2020; Li et al., 2021; Luo et al., 2023; Liu et al., 2023a; Yang et al., 2024c).

However, as today's model becomes larger and larger (e.g., the number of parameters of a state-of-the-art transformer-based models ranges from tens to hundreds of billions (Devlin et al., 2019; Dosovitskiy et al., 2020; Huang & Shu, 2024), a notable limitation of existing PFL methods is their dependence on multiple communication rounds to update models while in each round the amount of data that needs to be communicated is massive. While iterative communication is essential for progressively refining personalized models and improving accuracy, it becomes impracticable in many real-world scenarios where communication opportunities are expensive, limited, or infeasible. For instance, in Low Earth Orbit (LEO) satellite constellations, such as Starlink (SpaceX, 2020), OneWeb (OneWeb, 2021), and Kuiper (Amazon, 2022), satellites are strategically deployed in different orbits for distinct observational tasks (e.g., some predominantly monitor polar regions, while others focus on oceans or landmasses) (Doe & Smith, 2019). This specialized deployment makes multi-round communication with the same peer satellites untenable due to their high mobility and limited contact windows (Brown & Green, 2021). A

similar challenge is observed in autonomous vehicles operating in isolated regions with limited connectivity (Lee & Park, 2020). These vehicles may only intermittently connect to each other, making frequent communication impractical. Other real-world examples include mobile devices in areas with poor network coverage (Taylor & Nguyen, 2018) and Internet of Things (IoT) devices with sporadic internet access (Kim & Chen, 2019). Therefore, how to develop *one-shot* PFL (or OPFL in short) methods that can deliver *personalized* models with single-round communication (ideally, when two clients meet), so as to ensure superior and robust performance across non-IID clients' data remains a critical challenge for practical PFL implementation.

To address the challenges of communication constraints in federated learning, one-shot Federated Learning (OFL) has emerged as a promising paradigm, enabling the training of a global model with a single communication round between clients and the central server (Konečný et al., 2016; McMahan et al., 2017; Guha et al., 2019; Li et al., 2020a; Dai et al., 2024). Existing OFL methods primarily focus on two key strategies: ensemble refinement and synthesized data enhancement. Ensemble refinement focuses on improving the aggregation of local models into a better generalized global ensemble across clients by leveraging techniques such as parameter alignment, neuron matching, and weighted model combination (Su et al., 2023b; Liu et al., 2023b; Tang et al., 2024; Dai et al., 2024). On the other hand, synthesized data enhancement strategies aim to create synthesized data on the server side to facilitate global model training without accessing raw client data. This can involve dataset distillation, generative-based models, or data-free synthesis approaches (Zhou et al., 2020; Zhang et al., 2022a; Kang et al., 2023; Heinbaugh et al., 2023; Yang et al., 2024a;b).

At first glance, one might tend to assume that it is trivial/straightforward to obtain an OPFL model by directly fine-tuning the global model produced by OFL on each client's local dataset. However, as already shown in the large body of PFL literature, this belief does not hold in practice, especially for clients of highly diverse datasets. This is because the global model produced by an FL algorithm typically lacks the adaptive modules necessary for effective local adaptation. Without additional adaptation steps that explicitly designed for personalization, a direct fine-tuning approach tends to preserve the model's initial global biases instead of aligning it with each client's specific data distribution and needs. As a result, a local model fine-tuned from the global model over local dataset could perform arbitrarily poorly (Zhang et al., 2022b; Collins et al., 2022; Song et al., 2024).

Additionally, some practical considerations in the aforementioned real-world applications may also limit the applicability of building personalized models over a global model.

For example, it is not uncommon for a client to keep collecting new data during its operation, and periodically use its updated dataset to improve its personalized local model (see a satellite-based example in (Walden et al., 2020; Maskey & Cho, 2020)). The reliance of personalized models on the global model in this scenario requires periodic one-shot FL to update the global model, and hence the one-shot benefit essentially vanishes with time. Even worse, for those applications of an ad-hoc nature, e.g., an autonomous vehicle network, there does not exist an central parameter server, so none of the existing OFL-based methods is even feasible. Clearly, an OPFL method that does not rely on global model is highly desirable.

This paper proposes FOL (Federated Oriented Learning), a novel distributed OPFL method that allows a client to continuously improve its local model by learning from each of its neighbors through one-shot communication of their local models (e.g., in a LEO satellite network, this occurs when two satellites fly to each other's proximity, at which point communication cost via a direct inter-satellite link is lowest). While neighboring clients could have very diverse local datasets and hence very different local models, the tuning and pruning components in FOL allows a client to extract the most relevant portion of a neighbor's knowledge that is best aligned with the client's local model (and hence the word "oriented" in the name of the method), and then the ensemble component in FOL allows the client to integrate that portion into its local model, gaining strengthened representability for the specific local task of the client. In each round the client performs such knowledge extraction and integration for the top-$K$ neighbors that have the best knowledge alignment with the client, and hence strengthens the diversity of the client's local model. Benefiting from the shared knowledge between neighbors, the client's local model performs better than an isolatedly trained one over data unseen in its local training dataset. Two theoretical bounds on empirical risk discrepancy and convergence of the proposed FOL method are established. Extensive experiments were conducted on datasets Wildfire, Hurricane, CIFAR-10, CIFAR-100, and SVHN to demonstrate the effectiveness of FOL. Our experiment results verify that FOL consistently outperform counterparts, achieving accuracy improvements of up to 39.24% on Wildfire dataset.

## 2. Problem Statement

Given an image classification task, denote a set of $n$ clients as $\mathcal{K} = \{1, 2, \ldots, n\}$. Each client $k \in \mathcal{K}$ maintains a private dataset $\mathcal{D}^k = \{(x_i, y_i)\}_{i=1}^{|\mathcal{D}^k|}$, where $x_i$ is the input feature, $y_i$ is the corresponding label, and $|\mathcal{D}^k|$ is the dataset size. The goal of OPFL is to learn a personalized local model for each client $k$ by collecting a set of neighboring models through a one-shot communication process (each

pair of neighboring clients can exchange model parameters only once) and adapting these collected models to its own data distribution. For each model collection round $e \in \{1, \ldots, E\}$, let $\theta_k^{(e)-}$ and $\theta_k^{(e)+}$ be the parameters of client $k$'s local model at the beginning of the round and those of its updated local model after the learning, respectively, where $\theta_k^{(e)-} = \theta_k^{(e-1)+}$, i.e., it is carried over from the resulting model of the previous round. During round $e$, client $k$ collects models from its neighboring clients to form a model collection set $\{\phi_j^{(e)}\}_{j=1}^Q$, where $Q$ is a predefined number of other peers' models that each client can totally hold. Client $k$ then updates its own local model by integrating the collected models $\{\phi_j^{(e)}\}_{j=1}^Q$ with its current model $\theta_k^{(e-1)+}$ using its private dataset $\mathcal{D}^k$. The updated personalized model $\theta_k^{(e)+}$ is obtained by solving:

$$\theta_k^{(e)+} = \arg\min_\theta \quad \frac{1}{|\mathcal{D}^k|} \sum_{(x_i, y_i) \in \mathcal{D}^k} \ell(f_k(x_i; \theta | \theta_k^{(e)-}, \phi_1^{(e)}, ..., \phi_Q^{(e)}), y_i),$$
(1)

where $f_k(x_i; \theta)$ is $k$th client's prediction function that outputs the logits of $x_i$ given parameter $\theta$. $\ell$ is the cross entropy loss function. The updated model $\theta_k^{(e)+}$ is then used for subsequent rounds or tasks.

In contrast to OPFL, which explicitly addresses client-level personalization through local adaptation and/or refinement of received models, *traditional* OFL aims to learn a single global parameter $\theta_S$ that serves all clients uniformly. This approach can be formalized as:

$$\theta_S = \arg\min_{\theta_S} \quad \frac{1}{|\mathcal{D}_S|} \sum_{(x_i, y_i) \in \mathcal{D}_S} \ell(f_S(x_i; \theta_S), y_i), \quad (2)$$

where $\mathcal{D}_S$ represents public/synthesized data on the server side, and $f_S$ denotes the global model architecture.

## 3. Methodology

### 3.1. Architecture Overview

The proposed FOL framework is a multi-stage solution meticulously designed to solve the OPFL optimization problem articulated in Eq. (1). FOL systematically integrates a client's local model with a set of neighboring models through a sequence of coordinated stages, which ensures the derivation of an optimal personalized model $\theta_k^{(e)+}$ tailored to each client's unique data distribution. Specifically, FOL encompasses the following key stages:

Initially, each client $k \in \mathcal{K} = \{1, 2, \ldots, n\}$ trains an local model $\theta_k^{(1)-}$ on its local training dataset $\mathcal{D}_{\text{train}}^k$. This training process can be formalized as:

$$\theta_k^{(1)-} \leftarrow \arg\min_{\theta_k^0} \frac{1}{|\mathcal{D}_{\text{train}}^k|} \sum_{(x_i, y_i) \in \mathcal{D}_{\text{train}}^k} \ell(f_k(x_i; \theta_k^0), y_i). \quad (3)$$

After the initial preparation stage, each client starts to share its current local model with its peers. In each model collection round $e \in \{1, \ldots, E\}$, every client $k$ collects models

from its peer clients. Each client can maintain at most $Q$ peer models locally at any time. Once client $k$ has accumulated a total of $Q$ peer models, we denote this set as the model collection set $\{\phi_j^{(e)}\}_{j=1}^Q$. Each received neighboring model $\phi_j^{(e)}$ is then fine-tuned, pruned, and post fine-tuned on client $k$'s local training dataset $\mathcal{D}_{\text{train}}^k$, resulting in an adapted model $\phi_{j \to k}^{(e)}$. Afterward, these adapted neighbor models $\{\phi_{j \to k}^{(e)}\}_{j=1}^Q$ and the client's current local model, denoted by $\theta_k^{(e)-}$, are evaluated on a local validation set $\mathcal{D}_{\text{val}}^k$. The top-$K$ performing models are then selected based on their validation accuracy, forming the set $\{s_i^{(e)}\}_{i=1}^K$. These selected models are then combined into an ensemble by optimizing a weight vector $\mathbf{w}_k$ that reflects their relative contributions. Finally, knowledge distillation is employed to distill the ensemble's collective knowledge into an updated personalized model $\theta_k^{(e)+}$. This updated local model $\theta_k^{(e)+}$ is then carried over as client $k$'s initial model for the next round, i.e., $\theta_k^{(e+1)-} = \theta_k^{(e)+}$. The procedure of FOL is given in Algorithm 1.

### 3.2. Base Models Alignment

The collected models from neighboring clients may exhibit significant discrepancies with client $k$'s local data distribution. Such misalignments can lead to suboptimal ensemble performance and ineffective knowledge distillation. Therefore, it is crucial to ensure that the received neighboring models are well-aligned with the target client's local data distribution to maximize the benefits of subsequent ensemble and distillation steps. To address the aforementioned challenges, we propose a four-stage process.

**Fine-Tuning.** Upon receiving $\{\phi_j^{(e)}\}_{j=1}^Q$, client $k$ fine-tunes each of these models on its local training dataset $\mathcal{D}_{\text{train}}^k$ respectively to better align them with its local data distribution. This fine-tuning process can be formalized as:

$$\phi_{j \to k}^{'(e)} = \arg\min_\phi \frac{1}{|\mathcal{D}_{\text{train}}^k|} \sum_{(x_i, y_i) \in \mathcal{D}_{\text{train}}^k} \ell(f_j(x_i; \phi), y_i),$$
(4)

where $\phi$ is initialized by $\phi \leftarrow \phi_j^{(e)}$.

**Alignment-Aware Structured Pruning.** While fine-tuning helps to adapt a neighbor's model to local data, structured pruning can further ensures the model removes irrelevant or redundant filters/neurons for client $k$'s tasks (He & Xiao, 2023; Dery et al., 2024). In heterogeneous settings, neighboring models may have distinct architectures and specialized feature extractors that, if preserved appropriately, could benefit client $k$ in its future ensemble and distillation steps. However, traditional structured pruning methods (An et al., 2024; Sun & Shi, 2024) typically remove filters based solely on their local importance, without considering their alignment with the target model. This can lead to two key issues:

**Algorithm 1** Federated Oriented Learning (FOL)

1: **Input:** Set of clients $\mathcal{K}$, number of model collection rounds $E$, model collection size $Q$, top-$K$ selection size $K$, and local datasets $\mathcal{D}_{\text{train}}^k$ and $\mathcal{D}_{\text{val}}^k$.
2: **for** $e = 1$ to $E$ **do**
3:    **for** each client $k \in \mathcal{K}$ **do**
4:       **// Collect up to $Q$ neighbor models**
5:       $\{\phi_j^{(e)}\}_{j=1}^Q \leftarrow \text{CollectNeighbors}(k, \mathcal{K}, Q)$
6:       Initialize an empty list $\mathcal{S}_k^{(e)}$ to store scores
7:       **for** $j = 1$ to $Q$ **do**
8:          **// Fine-Tune neighbor model on client $k$**
9:          $\phi_{j\rightarrow k}^{'(e)} \leftarrow \text{FineTune}(\phi_j^{(e)}, \mathcal{D}_{\text{train}}^k)$
10:         **// Alignment-Based Structured Pruning**
11:         $\tilde{\phi}_{j\rightarrow k}^{(e)} \leftarrow \text{AlignmentPrune}(\phi_{j\rightarrow k}^{'(e)}, \theta_k^{(e)-}, \mathcal{D}_{\text{train}}^k)$
12:         **// Post-Pruning Fine-Tuning**
13:         $\phi_{j\rightarrow k}^{(e)} \leftarrow \text{PostFineTune}(\tilde{\phi}_{j\rightarrow k}^{(e)}, \mathcal{D}_{\text{train}}^k)$
14:         **// Evaluate on validation set**
15:         $\text{score}_k^{(e)}(\phi_{j\rightarrow k}^{(e)}) \leftarrow \text{Evaluate}(\phi_{j\rightarrow k}^{(e)}, \mathcal{D}_{\text{val}}^k)$
16:         Append $(\text{score}_k^{(e)}(\phi_{j\rightarrow k}^{(e)}), \phi_{j\rightarrow k}^{(e)})$ to $\mathcal{S}_k^{(e)}$
17:       **end for**
18:       **// Evaluate local model $\theta_k^{(e)-}$**
19:       $\text{score}_k^{(e)}(\theta_k^{(e)-}) \leftarrow \text{Evaluate}(\theta_k^{(e)-}, \mathcal{D}_{\text{val}}^k)$
20:       Append $(\text{score}_k^{(e)}(\theta_k^{(e)-}), \theta_k^{(e)-})$ to $\mathcal{S}_k^{(e)}$
21:       **// Select top-$K$ candidates by validation score, with cosine similarity as tie-breaker**
22:       $\{s_i^{(e)}\}_{i=1}^K \leftarrow \text{TopK}(\mathcal{S}_k^{(e)}, K,$
         $\text{CB} = \text{cosine\_similarity}(\theta_k^{(e)-}, \cdot))$
23:       **// Optimize ensemble weights**
24:       $\mathbf{w}_k^{(e)} \leftarrow \text{OptimizeWeights}(\{s_i^{(e)}\}_{i=1}^K, \mathcal{D}_{\text{train}}^k)$
25:       **// Distill knowledge into updated local model**
26:       $\theta_k^{(e)+} \leftarrow \text{Distill}(\theta_k^{(e)-}, \{s_i^{(e)}\}, \mathbf{w}_k^{(e)}, \mathcal{D}_{\text{train}}^k)$
27:    **end for**
28: **end for**
29: **Output:** $\{\theta_k^{(e)+}\}_{k\in\mathcal{K}}$

---

(1) over-pruning, where excessive removal of filters causes the received model to lose diverse feature extractors that could have been beneficial, and (2) excessive mismatch, where the retained filters extract features that are too different from those of the local model, making ensemble process ineffective due to highly inconsistent predictions.

To address this, we incorporate an alignment regularization term that guides the pruning process to retain filters in $\phi_{j\rightarrow k}^{'(e)}$ that are coherent with $\theta_k^{(e)-}$, thereby ensuring a structured adaptation rather than an arbitrary reduction of parameters.

Let: $\mathbb{L}_{\text{all}}^{(j\rightarrow k)}$ be the set of all layers in the neighbor model $\phi_{j\rightarrow k}^{'(e)}$. $\mathbb{L}_{\text{shared}} \subseteq \mathbb{L}_{\text{all}}^{(j\rightarrow k)}$ be the subset of layers that structurally match (i.e., have the same dimensionality and function type) with the local model $\theta_k^{(e)-}$. (The detailed definition of $\mathbb{L}_{\text{shared}}$ is given in Appendix B.1). $\mathbb{L}_{\text{unshared}} = \mathbb{L}_{\text{all}}^{(j\rightarrow k)} \setminus \mathbb{L}_{\text{shared}}$ be the layers unique to the neighbor model that have no direct counterpart in $\theta_k^{(e)-}$. For each shared layer $l \in \mathbb{L}_{\text{shared}}$, let $\mathbf{W}_l^{(j\rightarrow k)}$ and $\mathbf{W}_l^k$ denote the weight matrices of $\phi_{j\rightarrow k}^{'(e)}$ and $\theta_k^{(e)-}$, respectively. We introduce a gating vector $\boldsymbol{\alpha}_l = [\alpha_{l,1}, \ldots, \alpha_{l,m_l}]$ for each layer $l$, where $m_l$ is the number of filters (or neurons) in layer $l$. Each gating parameter $\alpha_{l,i} \in [0, 1]$ controls the retention or pruning of the $i$-th filter or neuron. For unshared layers $u \in \mathbb{L}_{\text{unshared}}$, we similarly define a gating vector $\boldsymbol{\alpha}_u$ to prune filters in that layer, but there is no alignment term because these layers do not correspond to any component in the local model. Hence, the pruned model $\tilde{\phi}_{j\rightarrow k}^{(e)}$ is obtained as the minimizer of the following objective function:

$$
\min_{\substack{\tilde{\phi}_{j\rightarrow k}^{(e)}, \\ \{\boldsymbol{\alpha}_l\}_{(l,l')\in\mathbb{L}_{\text{shared}}(k,j)}, \\ \{\boldsymbol{\alpha}_u\}_{u\in\mathbb{L}_{\text{unshared}}(k,j)}}} \underbrace{\frac{1}{|\mathcal{D}_{\text{train}}^k|} \sum_{(x_i,y_i)\in\mathcal{D}_{\text{train}}^k} \ell(f_j(x_i; \tilde{\phi}_{j\rightarrow k}^{(e)}, \{\boldsymbol{\alpha}_l\}, \{\boldsymbol{\alpha}_u\}), y_i)}_{\text{1. Task Loss on Local Data}}
$$

$$
+ \lambda_p \underbrace{\sum_{(l,l')\in\mathbb{L}_{\text{shared}}(k,j)} \sum_{i=1}^{m_l} \left\| \alpha_{l,i}\mathbf{W}_{l,i}^{(j\rightarrow k)} - \mathbf{W}_{l',i}^k \right\|_2^2}_{\text{2. Alignment Regularization (Shared Layers Only)}}
$$

$$
+ \gamma_{\text{shared}} \underbrace{\sum_{l\in\mathbb{L}_{\text{shared}}(k,j)} \left\| \boldsymbol{\alpha}_l \odot \mathbf{W}_l^{(j\rightarrow k)} \right\|_{2,1}}_{\text{3. Group-Lasso for Shared Layers}}
$$

$$
+ \gamma_{\text{unshared}} \underbrace{\sum_{u\in\mathbb{L}_{\text{unshared}}(k,j)} \left\| \boldsymbol{\alpha}_u \odot \mathbf{W}_u^{(j\rightarrow k)} \right\|_{2,1}}_{\text{4. Group-Lasso for Unshared Layers}},
$$

(5)

where $\lambda_p$ and $\gamma$ are hyperparameters controlling the strength of the alignment regularization and the structured pruning, respectively. $\|\cdot\|_{2,1}$ represents the group-lasso norm(Vogt & Roth, 2010). $\odot$ denotes element-wise multiplication.

**Post-Pruning Fine-Tuning** After the structured pruning, the pruned model $\tilde{\phi}_{j\rightarrow k}^{(e)}$ may experience a slight drop in performance due to the removal of certain filters or neurons. To recover any potential accuracy loss, we perform post fine-tuning:

$$
\phi_{j\rightarrow k}^{(e)} \leftarrow \arg\min_{\phi} \frac{1}{|\mathcal{D}_{\text{train}}^k|} \sum_{(x_i,y_i)\in\mathcal{D}_{\text{train}}^k} \ell(f_j(x_i; \phi), y_i), \quad (6)
$$

where $\phi$ is initialized by $\tilde{\phi}_{j\rightarrow k}^{(e)}$.

**Evaluation of Validation Accuracy.** For each pruned and post fine-tuned model $\phi_{j\rightarrow k}^{(e)}$ and the current local model $\theta_k^{(e)-}$, we compute the validation accuracy on the local validation set $\mathcal{D}_{\text{val}}^k$. The validation accuracy is defined as:

$$
\text{score}_k^{(e)}(\theta) = \frac{1}{|\mathcal{D}_{\text{val}}^k|} \sum_{(x_i,y_i)\in\mathcal{D}_{\text{val}}^k} \mathbb{1}(\arg\max f(x_i; \theta) = y_i),
$$

(7)

where $\mathbb{1}(\cdot)$ is the indicator function.
In scenarios where multiple models have the same validation

score, we employ cosine similarity to determine the best-aligned models relative to the local model $\theta_k^{(e)-}$. The cosine similarity between two models $\theta_1$ and $\theta_2$ is computed as:

$$\text{cosine\_similarity}(\theta_1, \theta_2) = \frac{\langle \theta_1, \theta_2 \rangle}{\|\theta_1\| \cdot \|\theta_2\|}, \tag{8}$$

where $\langle \theta_1, \theta_2 \rangle$ denotes the dot product of the parameter vectors of models $\theta_1$ and $\theta_2$. $\|.\|$ denotes the Euclidean norm. To ensure that only the most effective models contribute to the ensemble, client $k$ then selects the top-$K$ performing models based on these scores:

$$\{s_i^{(e)}\}_{i=1}^K = \text{TopK}(CB, \{\phi_{j \to k}^{(e)}\}_{j=1}^Q \cup \{\theta_k^{(e)-}\},$$
$$\{\text{score}_k^{(e)}(\phi_{j \to k}^{(e)})\}_{j=1}^Q \cup \{\text{score}_k^{(e)}(\theta_k^{(e)-})\}, K), \tag{9}$$

where $CB$ specifies that in the event of tied scores, models are further ranked using their cosine similarity to the local model $\theta_k^{(e)-}$, and pseudocode of TopK(.) is given in Algorithm 2.

---

**Algorithm 2** Top-$K$ Model Selection

---

**Require:** $\{\phi_{j \to k}^{(e)}\}_{j=1}^Q$, $\theta_k^{(e)-}$, $\mathcal{D}_{\text{val}}^k$, $K$
**Ensure:** $\{s_i^{(e)}\}_{i=1}^K$

1: Initialize an empty list $\mathcal{M}$
2: **for** each model $\phi_{j \to k}^{(e)}$ in $\{\phi_{j \to k}^{(e)}\}_{j=1}^Q$ **do**
3:     Compute $\text{score}_k^{(e)}(\phi_{j \to k}^{(e)})$ using Eq. equation 7
4:     Append $(\phi_{j \to k}^{(e)}, \text{score}_k^{(e)}(\phi_{j \to k}^{(e)}))$ to $\mathcal{M}$
5: **end for**
6: Compute $\text{score}_k^{(e)}(\theta_k^{(e)-})$ using Eq. equation 7
7: Append $(\theta_k^{(e)-}, \text{score}_k^{(e)}(\theta_k^{(e)-}))$ to $\mathcal{M}$
8: Sort $\mathcal{M}$ in descending order by validation score
9: **// Determine cutoff score for Tie-break**
10: Let $s^* =$ score of the $K$-th entry in $\mathcal{M}$
11: **// Split into Above and Tie group**
12: $\mathcal{M}_> \leftarrow \{v \in \mathcal{M} \mid \text{score}_k^{(e)}(v) > s^*\}$
13: $\mathcal{M}_= \leftarrow \{v \in \mathcal{M} \mid \text{score}_k^{(e)}(v) = s^*\}$
14: **// Start with all Above models**
15: $\mathcal{T} \leftarrow \mathcal{M}_>$
16: **// Tie-break only the cutoff group**
17: **for** each model $v \in \mathcal{M}_=$ **do**
18:     Compute cosine similarity between each model in the group and $\theta_k^{(e)-}$ using Eq. equation 8
19: **end for**
20: Sort the Tie group in descending order based on cosine similarity
21: Append the first $(K - |\mathcal{T}|)$ entries of $\mathcal{M}_=$ to $\mathcal{T}$
22: $\{s_i^{(e)}\}_{i=1}^K \leftarrow \mathcal{T}$
23: **Output:** $\{s_i^{(e)}\}_{i=1}^K$

---

### 3.3. Enhancing Ensemble Model Quality

The ensemble model functions as a "teacher" by aggregating knowledge from all selected base models $\{s_i^{(e)}\}_{i=1}^K$.

A prevalent approach to obtaining a global model is to directly average the parameters of individual models using FedAvg (McMahan et al., 2017). However, this method can be ineffective in scenarios with non-IID data distributions or significant heterogeneity in client model architectures (Li et al., 2024). Previous studies, such as (Zhang et al., 2022a), have attempted to address client models heterogeneity by employing simple or weighted averaging strategies, such as assigning uniform weights $w_j = \frac{1}{n}$ or weights proportional to client data sizes $w_j = \frac{n_k}{\sum_{k=1}^n n_k}$. However, as highlighted by (Wang et al., 2023), these averaging techniques are often ineffective in highly non-IID settings. To enhance ensemble performance, recent methods (Gong et al., 2021; Diao et al., 2023) typically require modifying each client's local training phase or transmitting additional information (e.g., intermediate gradients, model updates, or feature representations). While these approaches can be effective in many settings, they are often impractical in OFL settings due to increased communication overhead. In this study, we propose constructing a more effective weighted ensemble to address both client data and model heterogeneity, ensuring robust performance without incurring additional communication costs or altering local training processes.

**Optimal Weighted Ensemble.** To better accommodate heterogeneous data distributions and models, we introduce a weighted ensemble approach. Let $\{s_i^{(e)}\}_{i=1}^K$ denote the set of selected base models for client $k$ in round $e$. Our ensemble model $A_{\mathbf{w}_k^{(e)}}$ is formulated as:

$$A_{\mathbf{w}_k^{(e)}}(x; \{s_i^{(e)}\}_{i=1}^K) = \sum_{i=1}^K w_i^{(e)} \cdot f_i(x; s_i^{(e)}), \tag{10}$$

where $x$ is an input sample (e.g., an image), and $\mathbf{w}_k^{(e)} = \{w_i^{(e)}\}_{i=1}^K$ is a weight vector representing each model's relative contribution. Since the ensemble operation is applied at the logit layer, this reweighting mechanism can be seamlessly integrated into settings with heterogeneous or homogeneous model architectures.

To determine the optimal weights, client $k$ refines the weight vector $\mathbf{w}_k^{(e)}$ by solving the following optimization problem using its local training set $\mathcal{D}_{\text{train}}^k$:

$$\mathbf{w}_k^{(e)} = \arg\min_{\mathbf{w}_k^0} \frac{1}{|\mathcal{D}_{\text{train}}^k|} \sum_{(x_i, y_i) \in \mathcal{D}_{\text{train}}^k} \ell(A_{\mathbf{w}_k^0}(x_i; \{s_i^{(e)}\}_{i=1}^K), y_i). \tag{11}$$

### 3.4. Regularization-Based Knowledge Distillation

The ensemble outcome (a.k.a. the teacher model) is $K$-times larger than $\theta_k^{(e)-}$ in its size and therefore cannot be directly carried over to the next round. Instead, ideally, we need a smaller student model that captures the knowledge of the teacher mother and meanwhile is comparable to $\theta_k^{(e)-}$ in its *structure* and size.

To address this, we propose a *regularization-based knowledge distillation (KD)* mechanism that *(i)* matches the local

model's logits to those of the weighted ensemble (teacher) and *(ii)* constrains the student's parameters to remain close to the pre-distillation model $\theta_k^{(e)-}$, thereby preserving client-specific features and limiting the norm of weight updates. This distillation gives birth to $\theta_k^{(e)+}$ by minimizing the following KL-based distillation loss:

$$\mathcal{L}_{\text{KD}}(\theta_k^{(e)+}) = \frac{1}{|\mathcal{D}_{\text{train}}^k|} \sum_{x_i \in \mathcal{D}_{\text{train}}^k} \text{KL}\Big(\text{softmax}\Big(\frac{A_{\mathbf{w}_k^{(e)}}(x_i)}{T}\Big) \|$$

$$\text{softmax}\Big(\frac{f_k(x_i;\theta_k^{(e)+})}{T}\Big)\Big) + \lambda \|\theta_k^{(e)+} - \theta_k^{(e)-}\|^2, \quad (12)$$

where $\text{KL}(.\|.)$ denotes the Kullback–Leibler divergence between the teacher's output probability vector (softmax outputs) and the student's probability vector. The temperature parameter $T > 0$ controls the smoothness of the softmax distributions applied to the logits. $\lambda \geq 0$ weights the penalty on deviations of $\theta_k^{(e)+}$ from $\theta_k^{(e)-}$.

### 3.5. Theoretical Analysis

**A. Bound on the Empirical Risk Discrepancy Between Student and Teacher Models.** Without loss of generality, we analyze a single model collection round $e$ and omit the superscripts $+$ and $-$ for $\theta_k^{(e)}$ to streamline notation. We denote $\theta_k^{(e)}$ as the final updated local model for client $k$ at round $e$. The teacher ensemble's empirical risk on $\mathcal{D}_{\text{train}}^k$ is defined as:

$$R(A_{\mathbf{w}_k^{(e)}}) = \mathbb{E}_{(x_i,y_i) \sim \mathcal{D}_{\text{train}}^k}\Big[\ell(A_{\mathbf{w}_k^{(e)}}(x_i; \{s_i^{(e)}\}_{i=1}^K), y_i)\Big] \quad (13)$$

After solving the above distillation objective Eq. (12), the resulting student model $\theta_k^{(e)}$ has an empirical risk defined as:

$$R_{\text{S}}(\theta_k^{(e)}) = \mathbb{E}_{(x_i,y_i) \sim \mathcal{D}_{\text{train}}^k}\Big[\ell(f_k(x_i;\theta_k^{(e)}), y)\Big]. \quad (14)$$

The following theorem provides a quantitative bound on how closely the student model's empirical risk can align with that of the teacher ensemble.

**Theorem 1.** *Suppose the loss function $\ell(\hat{y}, y)$ (e.g., cross-entropy loss) is L-Lipschitz continuous with respect to the logit vector $\hat{y}$. Given a $C$-class classification task, the softmax outputs lie within the interval $(\alpha, 1 - \alpha)$ for some $0 < \alpha < 1$ across $C$ classes, and a temperature parameter $T > 0$, let the student model $\theta_k^{(e)}$ be trained to minimize the knowledge distillation loss $\mathcal{L}_{\text{KD}}(\theta_k^{(e)})$ as defined in Eq. 12. Then, the empirical risk discrepancy between the student and teacher models is bounded as follows:*

$$|R_{\text{S}}(\theta_k^{(e)}) - R(A_{\mathbf{w}_k^{(e)}})| \leq \frac{L \cdot C T}{\alpha(1-\alpha)} \cdot \Big(\frac{\mathcal{L}_{\text{KD}}(\theta_k^{(e)})}{2} + \frac{1}{8}\Big). \quad (15)$$

The proof of Theorem 1 is provided in Appendix B.2. Note that the assumption that the cross-entropy loss function

$\ell(\hat{y}, y)$ is L-Lipschitz continuous is widely adopted in ML literature to facilitate theoretical analyses and derive performance bounds (Mao et al., 2023; Safaryan et al., 2023).

**B. Convergence of Distillation.** Without loss of generality, we analyze a single model-collection round $e$ and omit the superscripts $+$ and $-$ for $\theta_k^{(e)}$ to streamline notation. We denote $\theta_k^r$ as the model parameters at local training epoch $r \in \{0, 1, \dots, R\}$. Let $\mathcal{L}_{\text{KD},k}(\theta)$, as defined in Eq. (12), represent the local distillation objective for client $k$, which measures the discrepancy between the student's output and the teacher's output. During each local epoch $r$, the client updates its parameters $\theta_k^r$ by:

$$\theta_k^{r+1} = \theta_k^r - \eta \nabla \mathcal{L}_{\text{KD},k}(\theta_k^r, \xi_k^r), \quad (16)$$

where $\eta > 0$ is the learning rate, and $\xi$ encapsulates any randomness (e.g., from mini-batch sampling).

To make the convergence analysis mathematically tractable, we follow standard practice (Regatti et al., 2022; Deng et al., 2024) and impose the following assumptions:

**Assumption 1.** The functions $\mathcal{L}_{\text{KD},k}(\cdot)$ for $k \in [K]$ are $L_s$-smooth. For all $\theta, \theta' \in \mathbb{R}^p$ (where $p$ denotes the total number of model parameters),

$$\|\nabla \mathcal{L}_{\text{KD},k}(\theta) - \nabla \mathcal{L}_{\text{KD},k}(\theta')\| \leq L_s \|\theta - \theta'\|. \quad (17)$$

**Assumption 2.** The function $\mathcal{L}_{\text{KD},k}(\cdot)$ is $\mu$-strongly convex; i.e., there exists $\mu > 0$ such that for all $\theta, \theta' \in \mathbb{R}^p$:

$$\mathcal{L}_{\text{KD},k}(\theta') \geq \mathcal{L}_{\text{KD},k}(\theta) + \langle \nabla \mathcal{L}_{\text{KD},k}(\theta), \theta' - \theta \rangle + \frac{\mu}{2} \|\theta' - \theta\|^2. \quad (18)$$

**Assumption 3.** The expectation of the squared $\ell_2$-norm of the stochastic gradients is bounded; that is, for all $\theta \in \mathbb{R}^p$,

$$\mathbb{E}_\xi[\|\nabla \mathcal{L}_{\text{KD},k}(\theta, \xi)\|^2] \leq \|\nabla \mathcal{L}_{\text{KD},k}(\theta)\|^2 + \sigma^2, \quad (19)$$

where $\sigma^2 > 0$ is a constant bounding the additional variance arising from stochastic gradient estimates.

The following theorem provides a convergence guarantee for the distillation process under the above assumptions.

**Theorem 2.** *Suppose $\{\theta_k^r\}_{r=0}^R$ are generated by $\theta_k^{r+1} = \theta_k^r - \eta \nabla \mathcal{L}_{\text{KD},k}(\theta_k^r, \xi_k^r)$, under the above assumptions. Then for all $r \geq 0$ and $0 < \eta < \frac{1}{L_s}$, the following bound holds:*

$$\mathbb{E}[\|\theta_k^r - \theta_k^*\|^2] \leq \gamma^r \|\theta_k^0 - \theta_k^*\|^2 + \sum_{\tau=0}^{r-1} \gamma^\tau \beta, \quad (20)$$

*where $\gamma = \Big(1 - 2\eta\mu + \frac{L_s^3}{\mu}\eta^2\Big)$, $\beta = \eta^2\sigma^2$, and $\theta_k^*$ is the stationary point of the distillation objective $\mathcal{L}_{\text{KD},k}$.* We defer the proof of Theorems 2 to Appendix B.3.

## 4. Experiments

While Theorems 1 and 2 theoretically demonstrate that our proposed FOL framework enhances accuracy for each participant in the system, it is essential to quantify these accuracy gains empirically. To this end, we conduct extensive experiments to evaluate the effectiveness of FOL.

*Table 1.* Test accuracies (%) on Wildfire and Hurricane ($\psi = 0.7$), reported as mean ± std.

| Dataset | Wildfire | | | Hurricane | | |
|---|---|---|---|---|---|---|
| Satellite # | 13 | 28 | 48 | 35 | 32 | 44 |
| Methods | $\psi = 0.7$ | | | | | |
| Local | 94.23 ± 1.84 | 94.12 ± 1.80 | 90.53 ± 1.57 | 86.93 ± 1.56 | 87.34 ± 1.60 | 89.82 ± 1.82 |
| FOL-A (E=1) | 97.19 ± 1.53 | 97.16 ± 1.24 | 95.97 ± 1.55 | 95.34 ± 1.42 | 96.18 ± 1.02 | 97.61 ± 1.68 |
| FOL-A (E=2) | 97.50 ± 1.12 | 97.52 ± 1.17 | 97.33 ± 1.23 | 96.59 ± 1.76 | 96.97 ± 1.41 | 97.87 ± 1.22 |
| FOL-A (E=3) | **97.53 ± 0.76** | **97.70 ± 0.98** | **97.99 ± 0.93** | **96.90 ± 1.09** | **97.47 ± 1.11** | **98.20 ± 1.03** |
| FOL (E=1) | 94.94 ± 1.38 | 95.21 ± 1.32 | 91.26 ± 1.62 | 90.09 ± 1.55 | 89.87 ± 0.69 | 91.62 ± 0.58 |
| FOL (E=2) | 95.23 ± 1.35 | 95.57 ± 0.72 | 91.60 ± 1.29 | 91.23 ± 1.57 | 91.77 ± 0.83 | 95.21 ± 1.49 |
| FOL (E=3) | 96.32 ± 0.96 | 95.75 ± 1.39 | 91.95 ± 1.31 | 92.26 ± 1.05 | 92.41 ± 1.68 | 95.81 ± 1.88 |
| FOL-AN (E=1) | 94.38 ± 1.86 | 94.86 ± 1.67 | 91.28 ± 1.82 | 88.24 ± 1.82 | 91.14 ± 1.13 | 92.81 ± 1.10 |
| FOL-AN (E=2) | 95.63 ± 1.40 | 95.04 ± 1.43 | 93.29 ± 1.51 | 90.09 ± 0.64 | 92.47 ± 1.86 | 94.01 ± 1.70 |
| FOL-AN (E=3) | 95.94 ± 0.71 | 96.45 ± 0.65 | 95.97 ± 1.43 | 93.19 ± 1.23 | 93.04 ± 1.19 | 96.41 ± 1.26 |
| FOL-N (E=1) | 93.44 ± 1.68 | 94.68 ± 1.79 | 88.59 ± 2.31 | 85.76 ± 1.85 | 89.22 ± 0.93 | 91.62 ± 1.19 |
| FOL-N (E=2) | 94.69 ± 0.53 | 94.86 ± 0.88 | 90.60 ± 1.01 | 89.16 ± 1.31 | 90.21 ± 1.28 | 92.22 ± 1.65 |
| FOL-N (E=3) | 95.31 ± 1.49 | 95.21 ± 0.98 | 91.95 ± 0.97 | 90.71 ± 0.59 | 90.51 ± 1.21 | 94.61 ± 0.73 |
| DENSE | 88.75 ± 1.91 | 87.41 ± 1.63 | 83.22 ± 1.57 | 67.49 ± 1.81 | 69.95 ± 1.70 | 73.05 ± 1.62 |
| Co-Boosting | 90.31 ± 1.26 | 89.19 ± 1.13 | 88.02 ± 1.25 | 72.14 ± 1.52 | 74.45 ± 1.72 | 74.04 ± 1.54 |
| FedAvg (E=1) | 73.19 ± 1.73 | 73.94 ± 1.96 | 68.18 ± 2.02 | 60.21 ± 1.73 | 62.03 ± 1.95 | 66.26 ± 1.62 |
| FedAvg (E=2) | 73.13 ± 1.91 | 72.29 ± 1.74 | 66.92 ± 1.55 | 59.44 ± 1.64 | 64.33 ± 1.33 | 69.88 ± 1.57 |
| FedAvg (E=3) | 74.61 ± 1.54 | 71.58 ± 1.16 | 68.48 ± 1.23 | 63.70 ± 0.71 | 65.16 ± 1.14 | 67.82 ± 0.92 |

*Table 2.* Test accuracies (%) on Wildfire and Hurricane ($\psi \in \{0.5, 0.3, 0.1\}$), reported as mean ± std.

| Dataset | Wildfire | | | Hurricane | | |
|---|---|---|---|---|---|---|
| Satellite # | 32 | 43 | 48 | 8 | 26 | 44 |
| Methods | $\psi = 0.5$ | $\psi = 0.3$ | $\psi = 0.1$ | $\psi = 0.5$ | $\psi = 0.3$ | $\psi = 0.1$ |
| Local | 79.07 ± 1.71 | 90.37 ± 1.76 | 85.50 ± 2.16 | 86.77 ± 1.90 | 57.14 ± 2.87 | 77.78 ± 1.35 |
| FOL-A (E=1) | 95.35 ± 1.42 | 94.07 ± 1.89 | 90.63 ± 1.92 | 95.04 ± 1.70 | 90.48 ± 1.57 | 88.89 ± 1.92 |
| FOL-A (E=2) | 96.52 ± 1.02 | 94.92 ± 1.25 | 96.14 ± 1.16 | 95.34 ± 1.16 | 91.72 ± 1.26 | 91.67 ± 1.26 |
| FOL-A (E=3) | **97.67 ± 0.71** | **95.76 ± 0.85** | **96.88 ± 1.01** | **95.87 ± 1.03** | **93.65 ± 1.14** | **94.44 ± 0.87** |
| FOL (E=1) | 90.70 ± 1.75 | 90.68 ± 1.01 | 88.46 ± 1.99 | 89.26 ± 1.25 | 84.13 ± 1.57 | 83.33 ± 1.69 |
| FOL (E=2) | 91.96 ± 1.09 | 91.53 ± 1.78 | 90.63 ± 1.77 | 90.08 ± 1.74 | 85.71 ± 1.38 | 84.43 ± 1.92 |
| FOL (E=3) | 93.02 ± 1.22 | 92.37 ± 1.27 | 93.75 ± 1.40 | 90.91 ± 1.38 | 87.30 ± 1.07 | 86.11 ± 1.18 |
| FOL-AN (E=1) | 90.77 ± 1.38 | 91.53 ± 1.26 | 87.51 ± 2.32 | 91.34 ± 1.70 | 87.47 ± 2.55 | 86.73 ± 1.94 |
| FOL-AN (E=2) | 93.22 ± 1.85 | 93.22 ± 1.17 | 90.63 ± 1.69 | 92.56 ± 1.18 | 88.89 ± 1.91 | 88.67 ± 1.75 |
| FOL-AN (E=3) | 95.35 ± 1.25 | 94.07 ± 1.21 | 90.94 ± 1.14 | 93.39 ± 1.37 | 90.48 ± 1.55 | 91.39 ± 1.26 |
| FOL-N (E=1) | 86.05 ± 1.96 | 88.14 ± 1.67 | 85.13 ± 1.92 | 85.95 ± 1.95 | 76.19 ± 1.73 | 80.56 ± 2.11 |
| FOL-N (E=2) | 87.35 ± 1.41 | 89.83 ± 1.76 | 86.38 ± 2.07 | 86.74 ± 1.83 | 80.95 ± 1.94 | 81.94 ± 1.38 |
| FOL-N (E=3) | 90.54 ± 1.51 | 90.06 ± 1.59 | 88.47 ± 1.37 | 87.60 ± 1.49 | 82.54 ± 1.76 | 83.37 ± 1.56 |
| DENSE | 79.91 ± 1.73 | 78.63 ± 1.98 | 52.08 ± 2.03 | 61.10 ± 1.51 | 58.73 ± 1.43 | 46.14 ± 1.81 |
| Co-Boosting | 86.05 ± 1.68 | 85.59 ± 1.65 | 54.51 ± 1.85 | 72.29 ± 1.68 | 52.38 ± 1.85 | 48.78 ± 1.50 |
| FedAvg (E=1) | 53.11 ± 1.82 | 63.25 ± 1.87 | 35.33 ± 2.76 | 66.12 ± 1.50 | 41.27 ± 1.99 | 46.14 ± 1.72 |
| FedAvg (E=2) | 56.03 ± 2.53 | 67.52 ± 1.92 | 45.16 ± 1.97 | 58.79 ± 1.86 | 45.16 ± 1.26 | 42.61 ± 1.86 |
| FedAvg (E=3) | 51.07 ± 1.93 | 66.10 ± 2.05 | 42.86 ± 1.53 | 60.33 ± 1.24 | 44.44 ± 1.76 | 43.33 ± 1.46 |

**Datasets.** We evaluate FOL's performance using four diverse datasets: Wildfire (Aaba, 2023), Hurricane (Park, 2021), CIFAR-10 (Krizhevsky, 2009), CIFAR-100 (Krizhevsky, 2009), and SVHN (Netzer et al., 2011). Specifically, the Wildfire dataset consists of 42,850 satellite images with a resolution of 350x350x3, designed for a binary classification task distinguishing wildfire events from non-fire regions. Similarly, the Hurricane dataset includes 14,000 satellite images with a resolution of 128x128x3, and it also focuses on a binary classification task that identifies hurricane weather patterns. CIFAR-10 and CIFAR-100 each provide 60,000 natural images ($32 \times 32 \times 3$) spanning 10 and 100 classes, respectively. Lastly, SVHN (Street View House Numbers) offers 73,257 training and 26,032 testing images ($32 \times 32 \times 3$) for 10-class digit recognition, sourced from real-world house number imagery. Please refer to Appendix C.3 for the summary of these datasets.

**Data partition.** In order to simulate the diverse local

datasets of LEO satellite constellations, we utilize a Dirichlet distribution-based approach to create non-IID data partitions across 70 clients (i.e., satellites). Specifically, we sample the proportion $p_k^i$ of data from each class $i$ allocated to client $k$ using a Dirichlet distribution $p_k \sim \text{Dir}(\psi)$, where $\psi$ controls the degree of data heterogeneity among clients. To enhance realism, we apply data augmentation techniques (e.g., rotation, flipping, cropping, resizing, normalization, and the addition of small Gaussian noise) for each image during the partitioning process. Using these augmentations, we mimic the variability introduced by different satellite perspectives. This ensures each client has unique data capturing the nuanced differences of real-world satellite imagery. Recognizing that newly launched LEO satellites, such as the Disaster Monitoring Constellation and OroraTech, typically need to collect several hundred to a thousand images before conducting model training and classification tasks (Walden et al., 2020; Maskey & Cho, 2020), we modify the partitioning function to guarantee that each client has a minimum dataset size of 100 images. Following the partitioning process, each client splits its local dataset into training, validation, and testing subsets in proportions of 70%, 15%, and 15%, respectively. This splitting is performed uniformly to maintain the same label distribution across all subsets, ensuring a rigorous evaluation of the personalization effectiveness of the FOL framework in handling non-IID data scenarios.

**Counterparts.** Given that each satellite has only a single opportunity to exchange models with the same neighbor, we compare the performance of FOL against three of the most relevant existing methods (baselines): FedAvg (McMahan et al., 2017), DENSE (Zhang et al., 2022a), and Co-Boosting (Dai et al., 2024). For FedAvg, we employ its standard configuration, where each satellite directly averages the parameters of all models received from its neighbors. The aggregated model is then evaluated on the respective satellite's local test dataset $\mathcal{D}_{\text{test}}^k$. Both DENSE and Co-Boosting are OFL-based methods. To ensure a convincing comparison, DENSE and Co-Boosting are evaluated using their final server models under their best performing configurations. Specifically, when the number of clients is set to 5, both the original reports and our experimental results (see Appendix C.4) indicate that these methods achieve their best performance. Consequently, we partition the dataset among 5 clients for these two method and train their final server models accordingly. Their performance is then evaluated on each satellite's respective local test dataset $\mathcal{D}_{\text{test}}^k$.

In addition, we investigate four FOL variants to highlight the impact of individual components. FOL represents the standard proposed framework, which combines fine-tuning, pruning, post-pruning fine-tuning, top-$K$ model selection, ensemble, and final knowledge distillation into a personalized model. FOL-A denotes the ensemble model. FOL-AN

and FOL-N omit the fine-tuning, pruning, and post-pruning fine-tuning stages but otherwise follow the same procedures as FOL-A and OFL, respectively.

**Model Configurations.** We consider a system with 70 satellites participating in $E = 3$ model collection rounds. Each satellite can store up to $Q = 29$ neighbor models at any point. The top-$K$ parameter is set to $K = 10$, meaning that after the first round (where 30 models are stored), each satellite retains the 10 highest-performing models locally and then gathers 20 additional models in the subsequent round, maintaining a total of 30. We adopt different hyperparameter configurations for different datasets. For the Wildfire and Hurricane datasets, we use Stochastic Gradient Descent (SGD) with a momentum of 0.9, a weight decay of 0.001, a learning rate of 0.001, a batch size of 32, a patience of 20, and local training for 200 epochs. For CIFAR-10, CIFAR-100, and SVHN, we use SGD with a momentum of 0.9, a weight decay of 0.001, a learning rate of 0.01, a batch size of 128, a patience of 20, and local training for 300 epochs. All baselines follow the same hyperparameter settings as ours. Results are reported as the average of 5 runs with different random seeds. All models are built in PyTorch and trained/tested on two GeForce RTX 4090 GPUs. Details on the network architectures are provided in Appendix C.5.

**Evaluation on Satellite Datasets.** Using traditional OFL evaluation metrics, which focus on average accuracy across all clients, is insufficient for assessing personalized OFL. This is because the primary goal of personalized OFL is to optimize performance for individual clients rather than to achieve a uniform global model. In personalized OFL, each client's data may be significantly different (i.e., heterogeneous), meaning average accuracy fails to capture the individualized improvements gained through personalization. Instead, evaluating a random subset of clients provides a more meaningful and realistic assessment, as it highlights how effectively a method adapts to diverse data distributions and addresses the distinct needs of each client. Ideally, we would conduct experiments for all 70 clients across each dataset setting. However, due to the constraints of our lab's computing resources, we are limited to a subset of experiments for each configuration. To be realistic, for $\psi = 0.7$ on the Wildfire and Hurricane datasets, we select three satellites: one is chosen randomly, and the other two are selected based on varying dataset sizes (ranging from a few hundred to a couple thousand images; see Appendix C.6). For other datasets and $\psi$ values, we randomly select one satellite to evaluate FOL's effectiveness, with the expectation that these results are representative of the general performance across the entire client population.

We begin by examining the Wildfire and Hurricane datasets under both moderate ($\psi = 0.7$) and high ($\psi \in \{0.5,$

$0.3, 0.1)$ levels of data heterogeneity. Table 1 presents the performance at $\psi = 0.7$, where the ensemble model FOL-A consistently achieves the highest accuracy during each model collection round. In addition, the final personalized model in FOL demonstrates substantial improvements over the best baseline, surpassing it by up to 6.56% on the Wildfire dataset (satellite #28) and by 21.77% on the Hurricane dataset (satellite #44). The larger improvement on the Hurricane dataset is likely due to its higher classification complexity, driven by intricate cloud formations, wind patterns, and other atmospheric phenomena, which challenge synthetic image generation methods like DENSE and Co-Boosting. These results highlight the ability of our FOL framework to enhance each satellite's model performance, which is especially beneficial for challenging classification tasks. Table 2 further explores the impact of increasing data skew by reducing $\psi$. While most methods exhibit performance drops under more pronounced non-IID conditions, FOL-A and FOL remain robust. Notably, FOL exceeds the best baseline by up to 39.24% on the Wildfire dataset (satellite #48) and 37.33% on the Wildfire dataset (satellite #44) when $\psi = 0.1$. These findings validate the effectiveness of our proposed FOL framework in non-IID satellite environments, consistently achieving significant improvements in both ensemble and personalized models.

*Table 3.* Test accuracies (%) on CIFAR-10, CIFAR-100, and SVHN, reported as mean ± std.

| Dataset | CIFAR-10 | CIFAR-100 | SVHN |
|---|---|---|---|
| Satellite # | 9 | 14 | 21 |
| Methods | $\psi = 0.7$ | $\psi = 0.7$ | $\psi = 0.5$ |
| Local | 60.06 ± 1.97 | 30.41 ± 2.28 | 78.97 ± 1.75 |
| FOL-A (E=1) | 70.73 ± 2.09 | 46.32 ± 2.12 | 85.73 ± 1.63 |
| FOL-A (E=2) | 70.93 ± 1.16 | 47.72 ± 1.53 | 86.26 ± 1.28 |
| FOL-A (E=3) | **71.02 ± 0.73** | **49.24 ± 1.12** | **88.37 ± 0.92** |
| FOL (E=1) | 65.68 ± 2.14 | 37.31 ± 2.43 | 81.09 ± 1.54 |
| FOL (E=2) | 66.06 ± 1.05 | 37.43 ± 1.58 | 81.62 ± 1.36 |
| FOL (E=3) | 66.83 ± 0.82 | 39.42 ± 1.03 | 82.85 ± 1.18 |
| FOL-AN (E=1) | 61.77 ± 2.36 | 31.23 ± 2.31 | 79.62 ± 1.64 |
| FOL-AN (E=2) | 62.35 ± 1.31 | 31.58 ± 1.75 | 80.92 ± 1.41 |
| FOL-AN (E=3) | 63.01 ± 0.71 | 32.05 ± 1.68 | 83.15 ± 1.33 |
| FOL-N (E=1) | 59.68 ± 2.43 | 30.64 ± 2.57 | 80.04 ± 1.74 |
| FOL-N (E=2) | 60.44 ± 1.37 | 30.99 ± 1.74 | 80.39 ± 1.39 |
| FOL-N (E=3) | 61.49 ± 1.43 | 31.11 ± 2.12 | 81.15 ± 1.22 |
| DENSE | 61.68 ± 2.03 | 29.59 ± 2.53 | 69.53 ± 1.57 |
| Co-Boosting | 63.11 ± 1.98 | 33.45 ± 2.12 | 73.58 ± 1.48 |
| FedAvg (E=1) | 47.76 ± 2.63 | 12.16 ± 3.93 | 53.08 ± 2.32 |
| FedAvg (E=2) | 44.90 ± 2.45 | 12.75 ± 3.23 | 58.72 ± 1.79 |
| FedAvg (E=3) | 45.57 ± 1.79 | 12.40 ± 2.89 | 55.49 ± 1.93 |

**Evaluation on CIFAR-10, CIFAR-100, and SVHN.** To illustrate the general applicability of our approach beyond satellite imaging classification tasks, we evaluate FOL on the CIFAR-10, CIFAR-100, and SVHN benchmarks using $\psi = 0.7$ (see Table 3). These datasets differ substantially from Wildfire and Hurricane in both task complexity

and data distribution. Nevertheless, FOL-A consistently achieves the highest accuracy, and FOL surpasses the best baseline by up to 3.72%, 5.97%, and 9.27% on CIFAR-10 (client #9), CIFAR-100 (client #14), and SVHN (client #21), respectively. These improvements highlight our framework's capability to adapt effectively across diverse domains, suggesting its potential applicability in a wide range of one-shot federated learning scenarios.

**Effects of the Proposed Components.** We further assess the contributions of our proposed model refinement process, including fine-tuning, structured pruning with alignment regularization, and post-pruning fine-tuning. Tables 1–3 demonstrate that FOL-A consistently outperforms FOL-AN in ensemble accuracy, while FOL consistently surpasses FOL-N in final personalized model accuracy. These substantial gains underscore the effectiveness of the proposed model refinement process in adapting received models to each client's unique data distribution, thereby significantly enhancing both model accuracy and robustness. Moreover, the performance of each FOL variant progressively improves as the number of model collection rounds $E$ increases (from 1 to 3). In each round, the updated local model $\theta_k^{(e)+}$ from the previous round serves as the initial local model for the next, enabling a gradual refinement process. As a result, the ensemble accuracy (FOL-A) steadily increases, while the final personalized model accuracy (FOL) also benefits from more effective knowledge distillation. This iterative enhancement ensures that each subsequent round begins with a stronger local model, leading to continuous performance gains across rounds. Overall, these results confirm that the integration of fine-tuning, structured pruning with alignment regularization, and post-pruning fine-tuning, along with ensemble-based knowledge distillation, is fundamental to the effectiveness of our FOL framework, particularly in handling non-IID satellite environments.

## 5. Conclusion

We introduced FOL, a distributed OPFL framework designed to handle communication constraints, high mobility, and diverse tasks in real-world environments. By integrating fine-tuning, pruning, post fine-tuning, ensemble refinement, and knowledge distillation, FOL enables clients to extract relevant knowledge from neighbors while aligning with their local models. We provide theoretical guarantees on the bounded risk discrepancy between the student and teacher models during knowledge distillation, as well as the convergence of the distillation process. Experiments on real-world datasets (Wildfire and Hurricane) and benchmarks (CIFAR-10, CIFAR-100, and SVHN) show that FOL consistently outperforms state-of-the-art one-shot FL methods, including FedAvg, DENSE, and Co-Boosting. Future work includes developing advanced personalization techniques and integrating stronger privacy guarantees into OPFL.

## Acknowledgments

This work is supported in part by the United States National Science Foundation (NSF) under grants CNS-2308761 and CNS-2006998. Any opinions, findings, conclusions, or recommendations expressed in this paper are those of the authors and do not necessarily reflect the views of the NSF.

## Impact Statement

This paper presents work whose goal is to advance the field of Machine Learning. There are many potential societal consequences of our work, none which we feel must be specifically highlighted here.

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

# A. Related Works

## A.1. One-Shot Federated Learning

One-Shot Federated Learning (OFL) has emerged as a promising paradigm that aims to train a global model using only a single communication round between clients and a central server (Konečnỳ et al., 2016; McMahan et al., 2017; Guha et al., 2019; Li et al., 2020a; Su et al., 2023b; Elmahallawy & Luo, 2023; Kang et al., 2023; Danilenka et al., 2023; Dai et al., 2024; Zeng et al., 2024; Yang et al., 2024b;a). Existing OFL methods primarily employ two key strategies: ensemble refinement and synthesized data enhancement. (1) Ensemble refinement aims to enhance the aggregation of local models into a more generalized global ensemble by utilizing techniques such as parameter alignment, neuron matching, and weighted model combination, including MA-Echo-OFL (Su et al., 2023b), FedLPA (Liu et al., 2023b), FuseFL (Tang et al., 2024), and Co-Boosting (Dai et al., 2024). (2) Synthesized data enhancement strategies, on the other hand, aim to generate surrogate or distilled data on the server side to facilitate global model training without direct access to raw client data. Methods in this category rely on dataset distillation, generative modeling, or data-free synthesis, such as DOSFL (Zhou et al., 2020), DENSE (Zhang et al., 2022a), FEDCVAE (Heinbaugh et al., 2023), FedISCA (Kang et al., 2023), LEOShot (Elmahallawy & Luo, 2023), IntactOFL (Zeng et al., 2024), FedDISC (Yang et al., 2024a), and FedDEO (Yang et al., 2024b).

While one-shot federated learning approaches significantly reduce communication overhead and are well-suited for resource-constrained environments, their effectiveness diminishes in highly heterogeneous settings where local data distributions vary substantially. This is because these methods prioritize generating a single global model that aims to perform generally across all clients. Consequently, the global-first paradigm restricts their applicability in scenarios requiring personalized federated learning, as a unified model cannot sufficiently address the specific needs of individual clients.

## A.2. Personalized Federated Learning

Personalized Federated Learning (PFL) has recently gained traction to address the limitations of a "one-size-fits-all" model in non-IID scenarios (Liang et al., 2020; Deng et al., 2020; T Dinh et al., 2020; Li et al., 2021; Oh et al., 2021; Collins et al., 2021; Tan et al., 2022b; Luo et al., 2023; Liu et al., 2023a; Su et al., 2023a; Wu et al., 2024; Yang et al., 2024c). PFL aims to customize each local model to perform well on client-specific data while still exploiting the shared knowledge from collaboration. Broadly, PFL methods can be categorized into two main groups. (1) Parameter decoupling approaches, which separate the model into a shared backbone (feature extractor) and a personalized head (classifier). The shared backbone captures universal features learned across the entire FL process, while the personalized head adapts to the specific data distribution of each client, including LG-FEDAVG (Liang et al., 2020), FedBABU (Oh et al., 2021), Fedproto (Tan et al., 2022b), PCCFED (Su et al., 2023a), and FedDecomp (Wu et al., 2024). (2) Optimization-based approaches, which leverage advanced techniques such as regularization methods, dynamic aggregation, and second-order optimization to constrain local models and adjust aggregation weights. This balance facilitates global collaboration while allowing for local personalization. Examples in this category include pFedMe (T Dinh et al., 2020), Ditto (Li et al., 2021), PGFed (Luo et al., 2023), Feddwa (Liu et al., 2023a), and FedAS (Yang et al., 2024c).

These PFL algorithms are particularly effective for clients with heterogeneous data distributions. To achieve strong performance in each client's local model, they rely on iterative updates across multiple communication rounds. However, this reliance poses significant challenges in time-sensitive or resource-constrained environments. For instance, in LEO satellite networks, where satellites have limited communication windows and high mobility, frequent model exchanges are impractical. Therefore, it is imperative to develop a practical one-shot PFL method that can effectively deliver high-quality personalized models within a single communication round.

# B. Supplementary Material on Methodology

## B.1. Definition of Shared and Unshared Layers

**Definition of Shared Layers.** Given a classification task, where each client maintains a neural network, let: $\mathbb{L}_k$ be the set of all layers in the local model $\theta_k^{(e)-}$. $\mathbb{L}_j$ be the set of all layers in the received neighbor model $\phi_{j\to k}^{'(e)}$. Specifically, $\theta_k^{(e)-}$ have layers $\{\text{Layer}_{k,1}, \text{Layer}_{k,2}, \ldots, \text{Layer}_{k,L_k}\}$, and $\phi_{j\to k}^{'(e)}$ have layers $\{\text{Layer}_{j,1}, \text{Layer}_{j,2}, \ldots, \text{Layer}_{j,L_j}\}$. We let each layer $\text{Layer}_{k,l} \in \mathbb{L}_k$ and $\text{Layer}_{j,l'} \in \mathbb{L}_j$. Then each layer $\text{Layer}_{k,l}$ or $\text{Layer}_{j,l'}$ can be characterized by both a *layer type* (e.g., convolution, fully connected) and a *dimensionality* or *shape* (e.g., number of filters, kernel size, input-output feature map dimensions). Formally, we let $\text{type}(\text{Layer}_{k,l})$ and $\text{dim}(\text{Layer}_{k,l})$ denote the layer type and dimensionality of $\text{Layer}_{k,l}$, respectively. Likewise for the neighbor model, we have $\text{type}(\text{Layer}_{j,l'})$ and $\text{dim}(\text{Layer}_{j,l'})$.

Then we define the set of shared layers, $\mathbb{L}_{\text{shared}}$, to be all *index pairs* $(l, l')$ such that:

1. The layers share the same type:

$$\text{type}(\text{Layer}_{k,l}) = \text{type}(\text{Layer}_{j,l'}), \tag{21}$$

2. Their dimensionalities match:

$$\text{dim}(\text{Layer}_{k,l}) = \text{dim}(\text{Layer}_{j,l'}). \tag{22}$$

Thus, we formally define:

$$\mathbb{L}_{\text{shared}}(k,j) = \left\{ (l,l') \in \mathbb{L}_k \times \mathbb{L}_j \mid \text{type}(\text{Layer}_{k,l}) = \text{type}(\text{Layer}_{j,l'}),\ \text{dim}(\text{Layer}_{k,l}) = \text{dim}(\text{Layer}_{j,l'}) \right\}, \tag{23}$$

where each $(l, l')$ pair is selected based on a first-available match to prevent multiple local layers from matching a single neighbor layer.

**Definition of Unshared Layers**. Any layer in $\phi_{j\to k}^{'(e)}$ that does not have a matching counterpart in $\theta_k^{(e)-}$ is considered unshared. We define:

$$\mathbb{L}_{\text{unshared}}(j \to k) = \mathbb{L}_j \setminus \left\{ l' \mid \exists l \in \mathbb{L}_k, (l,l') \in \mathbb{L}_{\text{shared}}(k,j) \right\}. \tag{24}$$

These layers have no direct local equivalent, meaning they must be pruned or retained based purely on their local utility, rather than structural alignment.

**Example: Identifying Shared & Unshared Layers:**

Consider two models:

1. Client $k$'s local CNN ($\theta_k^{(e)-}$):

$$\text{Conv}(3 \times 3, 32),\ \text{BN}(32), \text{Conv}(3 \times 3, 64),\ \text{FC}(256, 10)$$

2. Neighbor $j$'s received CNN ($\phi_{j\to k}^{'(e)}$):

$$\text{Conv}(3 \times 3, 32),\ \text{Conv}(3 \times 3, 64),\ \text{BN}(64),\ \text{FC}(256, 10)$$

Using our shared layer definition, we obtain:

$$\mathbb{L}_{\text{shared}} = \{(\text{Conv}(3 \times 3, 32)_k, \text{Conv}(3 \times 3, 32)_j), (\text{Conv}(3 \times 3, 64)_k, \text{Conv}(3 \times 3, 64)_j), (\text{FC}(256, 10)_k, \text{FC}(256, 10)_j)\}. \tag{25}$$

Thus, these layers form:

$$\mathbb{L}_{\text{unshared}} = \{\text{BN}(32)_j, \text{BN}(64)_j\}. \tag{26}$$

**B.2. Proof of Theorem 1**

**Proof.** We begin by clarifying how our *regularization-based* objective (i.e., KL + $\lambda$-penalty) still leads to the same KL bounds as the *pure* distillation loss. Specifically, recall the regularized objective:

$$\mathcal{L}_{\text{KD}}(\theta_k^{(e)}) \;=\; \frac{1}{|\mathcal{D}_k|} \sum_{x_i \in \mathcal{D}_k} \text{KL}\Big[\text{softmax}(\frac{A_{\mathbf{w}_k^{(e)}}(x_i)}{T}) \;\|\; \text{softmax}(\frac{f_k(x_i;\theta_k^{(e)})}{T})\Big] \;+\; \lambda\,\|\theta_k^{(e)+} - \theta_k^{(e)-}\|^2, \qquad (27)$$

where the second term is nonnegative. Hence, for any $\theta_k^{(e)}$,

$$\underbrace{\frac{1}{|\mathcal{D}_k|} \sum_{x_i \in \mathcal{D}_k} \text{KL}[\ldots]}_{\text{KL-only}} \;\leq\; \mathcal{L}_{\text{KD}}(\theta_k^{(e)}). \qquad (28)$$

**In other words, minimizing the entire (KL + regularization) objective also forces the pure KL portion to be small.** Consequently, *all subsequent bounding arguments (Pinsker's inequality, logit mismatch, etc.) remain valid under the regularized objective, since they only require that this KL portion be small.*

We now prove Theorem 1 in three main steps. First, we bound the discrepancy in empirical losses between the teacher and the student by showing that bounding *logit mismatch* (teacher vs. student logits) controls the cross-entropy discrepancy. Second, we link the logit mismatch to KL divergence via temperature scaling. Finally, we combine these steps to derive the risk discrepancy bound.

**Bounding the Empirical Loss Discrepancy via Logit Mismatch.** Let $\hat{z}_{\text{T}}(x_i) = \text{Logits}\Big(A_{\mathbf{w}_k}(x_i; \{s_i^{(e)}\}_{i=1}^K)\Big)$ denote the teacher ensemble's logits, and let $\hat{z}_{\text{S}}(x_i; \theta_k^{(e)}) = \text{Logits}(f_k(x_i; \theta_k^{(e)}))$ be the logits of the student model for the same input $x_i$. Since $\ell(\hat{z}, y_i)$ is assumed $L$-Lipschitz w.r.t. $\hat{z}$, we have, for any $\hat{z}_1, \hat{z}_2 \in \mathbb{R}^{y_d}$ and label $y_i$,

$$|\ell(\hat{z}_1, y_i) - \ell(\hat{z}_2, y_i)| \;\leq\; L\,\|\hat{z}_1 - \hat{z}_2\|. \qquad (29)$$

Hence, for each $(x_i, y_i) \in \mathcal{D}_{\text{train}}^k$,

$$|\ell(\hat{z}_{\text{S}}(x_i; \theta_k^{(e)}), y_i) - \ell(\hat{z}_{\text{T}}(x_i), y_i)| \;\leq\; L\,\|\hat{z}_{\text{S}}(x_i; \theta_k^{(e)}) - \hat{z}_{\text{T}}(x_i)\|. \qquad (30)$$

Taking expectation,

$$|R_{\text{S}}(\theta_k^{(e)}) - R(A_{\mathbf{w}_k})| \;\leq\; L \cdot \mathbb{E}_{(x_i, y_i) \in \mathcal{D}_{\text{train}}^k}\Big[\|\hat{z}_{\text{S}}(x_i; \theta_k^{(e)}) - \hat{z}_{\text{T}}(x_i)\|\Big]. \qquad (31)$$

Thus, bounding logit mismatch indeed bounds the cross-entropy discrepancy.

**Linking the KL Divergence to Logit Mismatch under Temperature Scaling.** Recall that in the *pure* KL-based distillation (temperature $T > 0$),

$$\mathcal{L}_{\text{KD}_{\text{pure}}}(\theta_k^{(e)}) = \frac{1}{|\mathcal{D}_{\text{train}}^k|} \sum_{x_i \in \mathcal{D}_{\text{train}}^k} \text{KL}\Big(\text{softmax}(\frac{\hat{z}_{\text{T}}(x_i)}{T}) \;\|\; \text{softmax}(\frac{\hat{z}_{\text{S}}(x_i; \theta_k^{(e)})}{T})\Big), \qquad (32)$$

and minimizing it makes the student's softened outputs $p_S$ close to the teacher's softened outputs $p_T$. (*Because of the nonnegative regularization in equation 12, minimising the* full *objective enforces at least the same KL closeness.*) We define

$$p_{\text{T}}(x_i) = \text{softmax}(\frac{\hat{z}_{\text{T}}(x_i)}{T}), \quad p_{\text{S}}(x_i; \theta_k^{(e)}) = \text{softmax}(\frac{\hat{z}_{\text{S}}(x_i; \theta_k^{(e)})}{T}). \qquad (33)$$

Hence, minimizing $\text{KL}(p_{\text{T}}(x_i)\|p_{\text{S}}(x_i; \theta_k^{(e)}))$ drives the student distribution $p_{\text{S}}(x_i; \theta_k^{(e)})$ to approximate the teacher distribution $p_{\text{T}}(x_i)$. Next, we aim to relate $\text{KL}(p_{\text{T}}(x_i)\|p_{\text{S}}(x_i; \theta_k^{(e)}))$ to the Euclidean distance between the logits $\|\hat{z}_{\text{S}}(x_i; \theta_k^{(e)}) - \hat{z}_{\text{T}}(x_i)\|_2$. To achieve this, we proceed through the following steps.

First, we work on bounding the distribution (softmax) discrepancy via logit mismatch. Consider the softmax function with temperature scaling:

$$p_c(z) = \frac{\exp(z_c)}{\sum_{j=1}^C \exp(z_j)}, \quad \text{where } z_c = \frac{\hat{z}_c}{T}, \quad c = 1, \ldots, C. \qquad (34)$$

Recall that the softmax outputs satisfy: $\alpha \leq p_c(z) \leq 1-\alpha$, for some $0 < \alpha < 1$. This is because if any class has a probability of exactly 1, all other class probabilities would need to be exactly 0, which is impossible since all exponentials $\exp(z) > 0$. In practice, especially with temperature scaling, the softmax outputs typically lie within the interval $[10^{-3}, 0.99]$, and thus a conservative bound on $\alpha$ is in the range $[10^{-3}, 10^{-2}]$.

Define the vector function $p : \mathbb{R}^C \to \mathbb{R}^C$ by:
$$p(z) = \text{softmax}(z).$$

The Jacobian matrix $J_p(z)$ of the softmax function is:

$$J_p(z) = \frac{\partial p}{\partial z} = \begin{bmatrix} \frac{\partial p_1}{\partial z_1} & \frac{\partial p_1}{\partial z_2} & \cdots & \frac{\partial p_1}{\partial z_C} \\ \frac{\partial p_2}{\partial z_1} & \frac{\partial p_2}{\partial z_2} & \cdots & \frac{\partial p_2}{\partial z_C} \\ \vdots & \vdots & \ddots & \vdots \\ \frac{\partial p_C}{\partial z_1} & \frac{\partial p_C}{\partial z_2} & \cdots & \frac{\partial p_C}{\partial z_C} \end{bmatrix} = \frac{1}{T} \left( \text{diag}(p(z)) - p(z)p(z)^\top \right), \tag{35}$$

where each element that insides the matrix is given by (according to (Bendersky, 2016)):

$$\frac{\partial p_c}{\partial z_d} = \begin{cases} \frac{1}{T} p_c(z)\left(1 - p_c(z)\right), & \text{if } c = d, \\ -\frac{1}{T} p_c(z)p_d(z), & \text{if } c \neq d. \end{cases} \tag{36}$$

Note that, the Jacobian matrix $J_p(z)$ of the softmax function is inherently rank-deficient, having a rank of $C-1$ due to the constraint $\sum_{c=1}^{C} p_c(z) = 1$. This rank deficiency poses a challenge for applying the Inverse Function Theorem directly, which is essential for linking the distribution discrepancy to the logit mismatch. To circumvent this issue, we will introduce a reduced softmax function.

Without loss of generality, we can fix the last one logit $p_C(z')$ (e.g., we can fix the last logit to 0.) to eliminate the redundancy introduced by the probability simplex constraint. Define the reduced softmax function as:

$$p(z') = (p_1(z'), p_2(z'), \ldots, p_{C-1}(z')), \tag{37}$$

where $z' = (\hat{z}_1, \hat{z}_2, \ldots, \hat{z}_{C-1})$ and $p_c(z')$ for $c = 1, \ldots, C-1$ are defined as:

$$p_c(z') = \frac{\exp(z'_c)}{1 + \sum_{j=1}^{C-1} \exp(z'_j)}. \tag{38}$$

Hence, the Jacobian matrix $J_p(z')$ of the reduced softmax function is:

$$J_p(z') = \frac{\partial p}{\partial z'} = \begin{bmatrix} \frac{\partial p_1}{\partial z'_1} & \frac{\partial p_1}{\partial z'_2} & \cdots & \frac{\partial p_1}{\partial z'_{C-1}} \\ \frac{\partial p_2}{\partial z'_1} & \frac{\partial p_2}{\partial z'_2} & \cdots & \frac{\partial p_2}{\partial z'_{C-1}} \\ \vdots & \vdots & \ddots & \vdots \\ \frac{\partial p_{C-1}}{\partial z'_1} & \frac{\partial p_{C-1}}{\partial z'_2} & \cdots & \frac{\partial p_{C-1}}{\partial z'_{C-1}} \end{bmatrix} = \frac{1}{T} \left( \text{diag}(p(z')) - p(z')(p(z'))^\top \right). \tag{39}$$

Note that $J_p(z')$ is full rank due to two reasons. (1). The diagonal elements satisfy $J_{cc} = p_c(z)(1 - p_c(z)) > 0$, so no row or column of the Jacobian matrix is entirely zero. (2). The off-diagonal elements of $J_f(z')$ satisfy $J_{cd} = -p_c(z)p_d(z), \quad c \neq d$. They do not induce linear dependencies between rows or columns because the probabilities $p_c(z)$ and $p_d(z)$ are strictly bounded by $\alpha$ and remain the probability simplex.

Since the reduced softmax function is composed of exponentials and sums, which are smooth and differentiable, and its Jacobian matrix is invertible. According to the Inverse Function Theorem(Wallach, 2005), the inverse Jacobian $J_{p^{-1}}(p)$ at $p = p(z'(x_i))$ can be defined as:
$$J_{p^{-1}}(p) = \left( J_p(z'(x_i)) \right)^{-1}. \tag{40}$$

Next, we need to bound the operator norm of the inverse Jacobian $J_{p^{-1}}(p)$. The operator norm of a matrix $A$, denoted $\|A\|_{2\to2}$, is defined as:
$$\|A\|_{2\to2} = \sup_{\|x\|_2=1} \|Ax\|_2. \tag{41}$$

For the inverse Jacobian $J_{p^{-1}}(p)$, according to (Birdal, 2019), the operator norm can be bounded in terms of the smallest singular value $\sigma_{\min}(J_p(z'(x_i)))$ of $J_p(z'(x_i))$:

$$\|J_{p^{-1}}(p')\|_{2\to2} = \frac{1}{\sigma_{\min}(J_p(z'(x_i)))}. \tag{42}$$

We then need to give the bound of $\sigma_{\min}(J_p(z'))$. Note that the full Jacobian and the reduced Jacobian coincide on the $(C-1)$-dimensional tangent space of the simplex, their *positive* singular values are identical. This is because the full Jacobian simply carries one extra singular value $0$ corresponding to the all-ones direction. Without loss generality, we simplify our notation in the following: Let $p = (p_1, \ldots, p_C) \in \mathbb{R}^C$ be a probability vector satisfying:

$$p_i \in (\alpha, 1-\alpha) \quad \text{for all } i, \quad \sum_{i=1}^{C} p_i = 1. \tag{43}$$

We then define the matrix $M$ as $M = \text{diag}(p) - pp^\top$. Since $\sum_{i=1}^{C} p_i = 1$, the matrix $M$ has rank at most $C-1$. Furthermore, we have:

$$M\mathbf{1} = \text{diag}(p)\mathbf{1} - p(p^\top\mathbf{1}) = p - p = \mathbf{0}. \tag{44}$$

Hence the vector $\mathbf{1} = (1, 1, \ldots, 1)$ spans the nullspace of $M$ with the eigenvalue $0$. Our goal is to give the bound for the minimum value of the positive nonzero eigenvalues of $M$. Formally, by the definition of Rayleigh quotient the minimum eigenvalues of $M$ can be defined as:

$$\lambda_{\min}^+(M) = \min_{\substack{x \neq 0 \\ x \perp \mathbf{1}}} \frac{x^\top M x}{x^\top x}, \tag{45}$$

where $x$ is a nonzero vector, and $x \perp \mathbf{1}$ can ensure the eigenvalue $\neq 0$.

Since the Rayleigh quotient $\dfrac{x^\top M x}{x^\top x}$ does not change if we scale $x$ by a nonzero constant. Hence, without loss of generality, we impose $\sum_{i=1}^{C} x_i^2 = 1$, i.e. $\|x\|_2 = 1$. Under this normalization,

$$\lambda_{\min}^+(M) = \min_{\substack{\|x\|_2=1 \\ \sum_i x_i=0}} \left[ x^\top M x \right]. \tag{46}$$

Next, we expand $x^\top M x$. Let $x = (x_1, \ldots, x_C)^\top \in \mathbb{R}^C$, we have:

$$x^\top M x = x^\top [\text{diag}(p) - pp^\top] x. \tag{47}$$

Split this into two terms:

$$\underbrace{x^\top \text{diag}(p)\, x}_{\textbf{Term A}} \quad \text{and} \quad \underbrace{x^\top [-pp^\top]\, x}_{\textbf{Term B}}. \tag{48}$$

**Term A: $x^\top \text{diag}(p)\, x$.** Since $\text{diag}(p)$ is diagonal with entries $\text{diag}(p)_{ii} = p_i$, we have

$$\text{diag}(p)\, x = (p_1 x_1,\ p_2 x_2,\ \ldots,\ p_C x_C)^\top, \tag{49}$$

Hence, we have:

$$x^\top [\text{diag}(p)]\, x = \sum_{i=1}^{C} (p_i x_i^2) = \sum_{i=1}^{C} p_i\, x_i^2. \tag{50}$$

**Term B: $x^\top [-pp^\top]\, x$.** Since $pp^\top$ is non-diagonal with entries $(pp^\top)_{ij} = p_i p_j$, we have:

$$(pp^\top) x = p(p^\top x) = \left( \sum_{k=1}^{C} p_k x_k \right) p. \tag{51}$$

Hence, we can get:

$$x^\top (p\,p^\top)\,x \;=\; \Big(\sum_{i=1}^{C} p_i x_i\Big)^2. \tag{52}$$

Add the negative sign:

$$x^\top [-\,p\,p^\top]\,x \;=\; -\Big(\sum_{i=1}^{C} p_i\,x_i\Big)^2. \tag{53}$$

Put eveything together, we get:

$$x^\top M\,x \;=\; \sum_{i=1}^{C} p_i\,x_i^2 \;-\; \Big(\sum_{i=1}^{C} p_i\,x_i\Big)^2. \tag{54}$$

To simplify this equation, let $z \;=\; \sum_{i=1}^{C} p_i x_i$, and since $\sum_i p_i = 1$, Eq. (54) can be rewritten as:

$$\sum_{i=1}^{C} p_i\,x_i^2 \;-\; z^2 \;=\; \sum_{i=1}^{C} p_i[x_i^2 - 2x_i z + z^2] \;-\; \Big(\sum_{i=1}^{C} p_i\Big) z^2 \;=\; \sum_{i=1}^{C} p_i(x_i - z)^2, \tag{55}$$

Now, the problem becomes to find the the global minimum of the following:

$$\sum_{i=1}^{C} p_i\,(x_i - z)^2 \quad \text{subject to} \quad \begin{cases} \sum_{i=1}^{C} x_i = 0, \\ \sum_{i=1}^{C} x_i^2 = 1, \\ p_i \in (\alpha,\,1-\alpha), \\ \sum_i p_i = 1, \end{cases} \tag{56}$$

Next, to solve this optimization problem, we follow (Yao et al., 2003), let $I_+ \subset \{1,\dots,C\}$ where $x_i = +a$ for $i \in I_+$, let $I_- \subset \{1,\dots,C\}$ where $x_i = -b$ for $i \in I_-$, and let $M = |I_+|$ and $N = |I_-|$, so that $M + N = C$. Hence, the constrains can be rewritten as:

$$\sum_{i=1}^{C} x_i = \underbrace{\sum_{i\in I_+} a}_{L\cdot a} + \underbrace{\sum_{i\in I_-} (-b)}_{N\cdot(-b)} \;=\; L\,a \;+\; N\,(-b) \;=\; L\,a \;-\; N\,b \;=\; 0 \quad \Longrightarrow \quad L\,a = N\,b. \tag{57}$$

Similarly, we have:

$$\sum_{i=1}^{C} x_i^2 \;=\; L\,a^2 \;+\; N\,b^2 \;=\; 1. \tag{58}$$

we define: $p_+ \;=\; \sum_{i\in I_+} p_i$, $p_- \;=\; \sum_{i\in I_-} p_i$ and $p_+ + p_- = 1$, where $p_+,\, p_- \in (\alpha,\,1-\alpha)$ because each $p_i \in (\alpha, 1-\alpha)$. The weighted mean $z$ can be rewritten as:

$$z \;=\; \sum_{i=1}^{C} p_i\,x_i \;=\; p_+\,(+a) \;+\; (p_-)\,(-b) \;=\; p_+\,a \;-\; p_-\,b. \tag{59}$$

The objective function can be rewritten as:

$$\sum_{i=1}^{C} p_i\,(x_i - z)^2 \;=\; p_+\,(a - z)^2 \;+\; p_-\,(-b - z)^2. \tag{60}$$

We then define the Lagrangian:

$$\mathcal{L}(a,b,p_+;\,\lambda,\mu) \;=\; p_+\,(a - z)^2 \;+\; (1 - p_+)\,(-b - z)^2 \;+\; \lambda\,(L\,a - N\,b) \;+\; \mu\,(L\,a^2 + N\,b^2 - 1), \tag{61}$$

where $z \;=\; p_+\,a \;-\; (1 - p_+)\,b \;=\; p_+\,a\ b \;+\; p_+\,b \;=\; p_+\,(a + b) \;-\; b$.

We then can solve this optimal function by taking partial derivatives $\frac{\partial \mathcal{L}}{\partial a} = 0$, $\frac{\partial \mathcal{L}}{\partial b} = 0$, $\frac{\partial \mathcal{L}}{\partial p_+} = 0$. The results gives $L = N$, $a = b$, $p_+ = \alpha$, $p_- = 1 - \alpha$.

By substituting everything back, we can get $z = \alpha(+a) + (1-\alpha)(-a) = a(\alpha - (1-\alpha)) = (2\alpha - 1)a$. Hence, we have $a - z = a - (2\alpha - 1)a = 2(1 - \alpha)a$, $-a - z = -a - (2\alpha - 1)a = -2\alpha a$.

Therefore, we have:

$$
\begin{aligned}
\sum_{i=1}^{C} p_i (x_i - z)^2 &= \alpha [a - z]^2 + (1 - \alpha)[-a - z]^2 \\
&= \alpha [2(1 - \alpha)a]^2 + (1 - \alpha)[-2\alpha a]^2 \\
&= 4a^2 \left[ \alpha (1 - \alpha)^2 + (1 - \alpha)\alpha^2 \right]. = \\
&= 4a^2 \alpha (1 - \alpha).
\end{aligned}
\tag{62}
$$

Recall that $L a^2 + N a^2 = C a^2 = 1$, thus $a^2 = 1/C$, so Eq. (62) can be rewritten as:

$$
\sum_{i=1}^{C} p_i (x_i - z)^2 = \frac{4}{C} \alpha (1 - \alpha).
\tag{63}
$$

By using the result of Eq. (63), include the parameter $T$, we get:

$$
\sigma_{\min}(J_p(z')) \geq \frac{4\alpha(1 - \alpha)}{CT},
\tag{64}
$$

We define:

$$
G_\alpha(T) = \frac{CT}{4\alpha(1 - \alpha)},
\tag{65}
$$

which serves as an upper bound for the operator norm of the inverse Jacobian.

Using the bounded inverse Jacobian, we then relate the differences in logits to the differences in the softmax outputs. Consider the student and teacher models with reduced softmax function for input $x_i$:

$$
p'_{\mathrm{S}}(x_i; \theta_k^{(e)}) = p(z'_{\mathrm{S}}(x_i; \theta_k^{(e)})), \quad p'_{\mathrm{T}}(x_i) = p(z'_{\mathrm{T}}(x_i)).
\tag{66}
$$

Hence, we have:

$$
z'_{\mathrm{S}}(x_i; \theta_k^{(e)}) - z'_{\mathrm{T}}(x_i) = p^{-1}(p'_{\mathrm{S}}(x_i; \theta_k^{(e)})) - p^{-1}(p'_{\mathrm{T}}(x_i)).
\tag{67}
$$

Applying the Mean Value Theorem for vector-valued functions, there exists a point $\xi$ on the line segment between $p'_{\mathrm{S}}(x_i; \theta_k^{(e)})$ and $p'_{\mathrm{T}}(x_i)$ such that:

$$
z'_{\mathrm{S}}(x_i; \theta_k^{(e)}) - z'_{\mathrm{T}}(x_i) \leq J_{p^{-1}}(\xi) \cdot (p'_{\mathrm{S}}(x_i; \theta_k^{(e)}) - p'_{\mathrm{T}}(x_i)).
\tag{68}
$$

Taking the $L_2$ norm on both sides:

$$
\|z'_{\mathrm{S}}(x_i; \theta_k^{(e)}) - z'_{\mathrm{T}}(x_i)\|_2 \leq \|J_{p^{-1}}(\xi)\|_{2 \to 2} \cdot \|p'_{\mathrm{S}}(x_i; \theta_k^{(e)}) - p'_{\mathrm{T}}(x_i)\|_2.
\tag{69}
$$

Substituting the bound on the inverse Jacobian:

$$
\|z'_{\mathrm{S}}(x_i; \theta_k^{(e)}) - z'_{\mathrm{T}}(x_i)\|_2 \leq G_\alpha(T) \cdot \|p'_{\mathrm{S}}(x_i; \theta_k^{(e)}) - p'_{\mathrm{T}}(x_i)\|_2.
\tag{70}
$$

We then extend the bound in Eq. (70) by relax the reduced softmax to the full softmax. First, we extend the LHS. Recall that $z_S$ and $z_T$ are the full $C$-dimensional logit vectors for the student and teacher models, respectively. The reduced logit vectors $z'_S$ and $z'_T$ exclude the last component (fixed to 0), i.e., $z'_S = (z_{S,1}, z_{S,2}, \ldots, z_{S,C-1})$ and similarly for $z'_T$. Since the last logit is fixed ($z_{S,C}, z_{T,C} = 0$), the Euclidean distance between the full logits is equal to the Euclidean distance between the reduced logits:

$$
\|z_S - z_T\|_2^2 = \sum_{c=1}^{C} (z_{S,c} - z_{T,c})^2 = \sum_{c=1}^{C-1} (z'_{S,c} - z'_{T,c})^2 + (z_{S,C} - z_{T,C})^2 = \|z'_S - z'_T\|_2^2 + 0
\tag{71}
$$

To extend the RHS of Eq. (70), recall that $p_S = \text{softmax}(z_S)$, $p_T = \text{softmax}(z_T)$, $p'_S = (p_{S,1}, p_{S,2}, \ldots, p_{S,C-1})$, $p'_T = (p_{T,1}, p_{T,2}, \ldots, p_{T,C-1})$, $p_{S,C} = 1 - \sum_{c=1}^{C-1} p_{S,c}$, and $p_{T,C} = 1 - \sum_{c=1}^{C-1} p_{T,c}$. The norm of full softmax distributions can be express as:

$$\|p_S - p_T\|_2^2 = \sum_{c=1}^{C-1} (p_{S,c} - p_{T,c})^2 + (p_{S,C} - p_{T,C})^2. \tag{72}$$

For the RHS of Eq. (72), we have:

$$p_{S,C} - p_{T,C} = \left(1 - \sum_{c=1}^{C-1} p_{S,c}\right) - \left(1 - \sum_{c=1}^{C-1} p_{T,c}\right) = \sum_{c=1}^{C-1} (p_{T,c} - p_{S,c}). \tag{73}$$

Plugging this back into Eq. (72), we have:

$$\|p_S - p_T\|_2^2 = \sum_{c=1}^{C-1} (p_{S,c} - p_{T,c})^2 + \left(\sum_{c=1}^{C-1} (p_{T,c} - p_{S,c})\right)^2. \tag{74}$$

Notice that $\left(\sum_{c=1}^{C-1} (p_{T,c} - p_{S,c})\right)^2 \geq 0$. Hence, we have:

$$\|p_S - p_T\|_2^2 \geq \sum_{c=1}^{C-1} (p_{S,c} - p_{T,c})^2 = \|p'_S - p'_T\|_2^2. \tag{75}$$

Therefore, the bound in Eq. (70) can be extended by:

$$\|z_S - z_T\|_2 = \|z'_S - z'_T\|_2 \leq G_\alpha(T) \cdot \|p'_S - p'_T\|_2 \leq G_\alpha(T) \cdot \|p_S - p_T\|_2. \tag{76}$$

To connect the KL divergence to the Euclidean distance between the softmax outputs, we utilize Pinsker's Inequality along with norm relationships:

$$\text{KL}(p_T(x_i)\|p_S(x_i; \theta_k^{(e)})) \geq \frac{1}{2}\|p_T(x_i) - p_S(x_i; \theta_k^{(e)})\|_1^2. \tag{77}$$

Additionally, in $\mathbb{R}^C$, the $L^1$ and $L^2$ norms satisfy:

$$\|p_T(x_i) - p_S(x_i; \theta_k^{(e)})\|_2 \leq \|p_T(x_i) - p_S(x_i; \theta_k^{(e)})\|_1. \tag{78}$$

**Derivation of** $\|p(z_1) - p(z_2)\|_2 \leq \|p(z_1) - p(z_2)\|_1$**:**

We start with the $L_1$-norm and $L_2$-norm definitions: $\|p(z_1) - p(z_2)\|_1 = \sum_{i=1}^C |p_i(z_1) - p_i(z_2)|$, $\|p(z_1) - p(z_2)\|_2^2 = \sum_{i=1}^C (p_i(z_1) - p_i(z_2))^2$. Then, we square of the $L_1$-norm:

$$\|p(z_1) - p(z_2)\|_1^2 = \left(\sum_{i=1}^C |p_i(z_1) - p_i(z_2)|\right)^2 = \sum_{i=1}^C |p_i(z_1) - p_i(z_2)| \sum_{j=1}^C |p_j(z_1) - p_j(z_2)|. \tag{79}$$

Eq. (79) can be decomposed as:

$$\sum_{i=1}^C |p_i(z_1) - p_i(z_2)| \sum_{j=1}^C |p_j(z_1) - p_j(z_2)| = \sum_{i=j}^C |p_i(z_1) - p_i(z_2)|^2 + \sum_{i \neq j} |p_i(z_1) - p_i(z_2)||p_j(z_1) - p_j(z_2)|. \tag{80}$$

This implies:

$$\|p(z_1) - p(z_2)\|_1^2 = \|p(z_1) - p(z_2)\|_2^2 + \sum_{i \neq j} |p_i(z_1) - p_i(z_2)||p_j(z_1) - p_j(z_2)|. \tag{81}$$

Therefore, we proofed the LHS of Eq. (78):

$$\|p(z_1) - p(z_2)\|_2^2 \leq \|p(z_1) - p(z_2)\|_1^2. \tag{76}$$

Combining Eq. (77) and Eq. (78), we obtain:

$$\|p_\mathrm{T}(x_i) - p_\mathrm{S}(x_i; \theta_k^{(e)})\|_2 \leq \sqrt{2 \cdot \mathrm{KL}(p_\mathrm{T}(x_i) \| p_\mathrm{S}(x_i; \theta_k^{(e)}))}. \tag{82}$$

**Establishing the Upper Bound on Logit Mismatch**

By combining Eq. (76) and Eq. (82), we can get the logit mismatch bound:

$$\|\hat{z}_\mathrm{S}(x_i; \theta_k^{(e)}) - \hat{z}_\mathrm{T}(x_i)\|_2 \leq G_\alpha(T) \cdot \sqrt{2 \cdot \mathrm{KL}(p_\mathrm{T}(x_i) \| p_\mathrm{S}(x_i; \theta_k^{(e)}))}. \tag{83}$$

To establish a linear relationship, we employ the inequality $\sqrt{x} \leq \frac{x}{2} + \frac{1}{2}$:

$$\sqrt{2 \cdot \mathrm{KL}(p_\mathrm{T}(x_i) \| p_\mathrm{S}(x_i; \theta_k^{(e)}))} \leq \frac{2 \cdot \mathrm{KL}(p_\mathrm{T}(x_i) \| p_\mathrm{S}(x_i; \theta_k^{(e)}))}{2} + \frac{1}{2}. \tag{84}$$

Substituting back:

$$\|\hat{z}_\mathrm{S}(x_i; \theta_k^{(e)}) - \hat{z}_\mathrm{T}(x_i)\|_2 \leq G_\alpha(T) \cdot \left( \frac{2 \cdot \mathrm{KL}(p_\mathrm{T}(x_i) \| p_\mathrm{S}(x_i; \theta_k^{(e)}))}{2} + \frac{1}{2} \right). \tag{85}$$

Hence, we have:

$$\|\hat{z}_\mathrm{S}(x_i; \theta_k^{(e)}) - \hat{z}_\mathrm{T}(x_i)\|_2 \leq G_\alpha(T) \cdot \mathrm{KL}(p_\mathrm{T}(x_i) \| p_\mathrm{S}(x_i; \theta_k^{(e)})) + \frac{G_\alpha(T)}{2}. \tag{86}$$

Define:

$$\kappa = 2 G_\alpha(T), \quad \delta_0 = \frac{G_\alpha(T)}{2}, \tag{87}$$

we obtain:

$$\|\hat{z}_\mathrm{S}(x_i; \theta_k^{(e)}) - \hat{z}_\mathrm{T}(x_i)\|_2 \leq \kappa \cdot \mathrm{KL}(p_\mathrm{T}(x_i) \| p_\mathrm{S}(x_i; \theta_k^{(e)})) + \delta_0. \tag{88}$$

**Deriving the final bound**

By averaging the above inequality over all samples in the training dataset $\mathcal{D}_\mathrm{train}^k$, we get:

$$\mathbb{E}_{(x_i, y_i) \in \mathcal{D}_\mathrm{train}^k} \left[ \|\hat{z}_\mathrm{S}(x_i; \theta_k^{(e)}) - \hat{z}_\mathrm{T}(x_i)\|_2 \right] \leq \kappa \cdot \mathcal{L}_{\mathrm{KD}_\mathrm{pure}}(\theta_k^{(e)}) + \delta_0, \tag{89}$$

where:

$$\mathcal{L}_{\mathrm{KD}_\mathrm{pure}}(\theta_k^{(e)}) = \frac{1}{|\mathcal{D}_\mathrm{train}^k|} \sum_{x_i \in \mathcal{D}_\mathrm{train}^k} \mathrm{KL}(p_\mathrm{T}(x_i) \| p_\mathrm{S}(x_i; \theta_k^{(e)})). \tag{90}$$

Substituting back the regularized objective $\mathcal{L}_\mathrm{KD}(\theta_k^{(e)})$, and by using Eq. (28) we have:

$$\mathbb{E}_{(x_i, y_i) \in \mathcal{D}_\mathrm{train}^k} \left[ \|\hat{z}_\mathrm{S}(x_i; \theta_k^{(e)}) - \hat{z}_\mathrm{T}(x_i)\|_2 \right] \leq \kappa \cdot \mathcal{L}_{\mathrm{KD}_\mathrm{pure}}(\theta_k^{(e)}) + \delta_0 \leq \kappa \cdot \mathcal{L}_\mathrm{KD}(\theta_k^{(e)}) + \delta_0, \tag{91}$$

By substituting back the $G_\alpha(T), \kappa$, and $\delta_0$, we have

$$\mathbb{E}_{(x_i, y_i) \in \mathcal{D}_\mathrm{train}^k} \left[ \|\hat{z}_\mathrm{S}(x_i; \theta_k^{(e)}) - \hat{z}_\mathrm{T}(x_i)\|_2 \right] \leq \frac{CT}{2\alpha(1-\alpha)} \cdot \mathcal{L}_\mathrm{KD}(\theta_k^{(e)}) + \frac{CT}{8\alpha(1-\alpha)}. \tag{92}$$

$$| R_\mathrm{S}(\theta_k^{(e)}) - R(A_{\mathbf{w}_k}) | \leq L \cdot \mathbb{E}_{(x_i, y_i) \in \mathcal{D}_\mathrm{train}^k} \left[ \|\hat{z}_\mathrm{S}(x_i; \theta_k^{(e)}) - \hat{z}_\mathrm{T}(x_i)\| \right]. \tag{93}$$

Let's then combine Eq. (93) and Eq. (31):

$$| R_\mathrm{S}(\theta_k^{(e)}) - R(A_{\mathbf{w}_k}) | \leq \frac{L \cdot CT}{2\alpha(1-\alpha)} \cdot \mathcal{L}_\mathrm{KD}(\theta_k^{(e)}) + \frac{L \cdot CT}{8\alpha(1-\alpha)}. \tag{94}$$

**Final Bound and Conclusion.** Combining the above mismatch and KL relationships, we obtain

$$| R_\mathrm{S}(\theta_k^{(e)}) - R(A_{\mathbf{w}_k}) | \leq \frac{L \cdot CT}{\alpha(1-\alpha)} \left[ \frac{\mathcal{L}_\mathrm{KD}(\theta_k^{(e)})}{2} + \frac{1}{8} \right]. \tag{95}$$

This completes the proof of Theorem 1. $\qquad\square$

### B.3. Proof of Theorem 2

**Proof.** In order to prove Theorem 2, we first need to expand the SGD update rule:

Recall the update rule:

$$\theta_k^{r+1} = \theta_k^r - \eta \nabla \mathcal{L}_{\text{KD},k}(\theta_k^r, \xi_k^r). \tag{96}$$

Since we want to bound $\|\theta_k^{r+1} - \theta_k^*\|^2$, let's expand the squared norm:

$$\|\theta_k^{r+1} - \theta_k^*\|^2 = \|\theta_k^r - \theta_k^* - \eta \nabla \mathcal{L}_{\text{KD},k}(\theta_k^r, \xi_k^r)\|^2. \tag{97}$$

Using $(u - v)^2 = \|u\|^2 - 2\langle u, v \rangle + \|v\|^2$, we have:

$$\|\theta_k^{r+1} - \theta_k^*\|^2 = \|\theta_k^r - \theta_k^*\|^2 - 2\eta\langle\theta_k^r - \theta_k^*, \nabla \mathcal{L}_{\text{KD},k}(\theta_k^r, \xi_k^r)\rangle + \eta^2 \|\nabla \mathcal{L}_{\text{KD},k}(\theta_k^r, \xi_k^r)\|^2. \tag{98}$$

We then take conditional expectation given $\theta_k^r$ on both side of Eq. (98),

$$\mathbb{E}[\|\theta_k^{r+1} - \theta_k^*\|^2 \mid \theta_k^r] = \|\theta_k^r - \theta_k^*\|^2 - 2\eta \underbrace{\mathbb{E}[\langle\theta_k^r - \theta_k^*, \nabla \mathcal{L}_{\text{KD},k}(\theta_k^r, \xi_k^r)\rangle]}_{\text{Term A}} + \eta^2 \underbrace{\mathbb{E}[\|\nabla \mathcal{L}_{\text{KD},k}(\theta_k^r, \xi_k^r)\|^2]}_{\text{Term B}}. \tag{99}$$

By the definition of the stochastic gradient (the expectation of the stochastic gradient equals the true gradient), we have:

$$\mathbb{E}[\nabla \mathcal{L}_{\text{KD},k}(\theta_k^r, \xi_k^r)] = \nabla \mathcal{L}_{\text{KD},k}(\theta_k^r). \tag{100}$$

Hence, Term A $= \langle\theta_k^r - \theta_k^*, \nabla \mathcal{L}_{\text{KD},k}(\theta_k^r)\rangle$.

The **derivation** for the Term A:

By the definition of the inner product for vectors, we have:

$$\langle\theta_k^r - \theta_k^*, \nabla \mathcal{L}_{\text{KD},k}(\theta_k^r, \xi_k^r)\rangle = \sum_{i=1}^{n}(\theta_k^r[i] - \theta_k^*[i]) \cdot \nabla \mathcal{L}_{\text{KD},k}(\theta_k^r, \xi_k^r)[i], \tag{101}$$

where $\theta_k^r[i]$, $\theta_k^*[i]$, and $\nabla \mathcal{L}_{\text{KD},k}(\theta_k^r, \xi_k^r)[i]$ are the $i$-th components of $\theta_k^r$, $\theta_k^*$, and the stochastic gradient, respectively.

We then take the expectation on both side of Eq. (101) over $\xi_k^r$,

$$\mathbb{E}[\langle\theta_k^r - \theta_k^*, \nabla \mathcal{L}_{\text{KD},k}(\theta_k^r, \xi_k^r)\rangle] = \mathbb{E}\left[\sum_{i=1}^{n}(\theta_k^r[i] - \theta_k^*[i]) \cdot \nabla \mathcal{L}_{\text{KD},k}(\theta_k^r, \xi_k^r)[i]\right]. \tag{102}$$

Since the expectation operator $\mathbb{E}$ is linear, so we can rearrange the above equation as:

$$\mathbb{E}[\langle\theta_k^r - \theta_k^*, \nabla \mathcal{L}_{\text{KD},k}(\theta_k^r, \xi_k^r)\rangle] = \sum_{i=1}^{n}(\theta_k^r[i] - \theta_k^*[i]) \cdot \mathbb{E}[\nabla \mathcal{L}_{\text{KD},k}(\theta_k^r, \xi_k^r)[i]]. \tag{103}$$

Substituting Eq. (100) into the above equation, we have:

$$\sum_{i=1}^{n}(\theta_k^r[i] - \theta_k^*[i]) \cdot \mathbb{E}[\nabla \mathcal{L}_{\text{KD},k}(\theta_k^r, \xi_k^r)[i]] = \sum_{i=1}^{n}(\theta_k^r[i] - \theta_k^*[i]) \cdot \nabla \mathcal{L}_{\text{KD},k}(\theta_k^r)[i]. \tag{104}$$

Note that the summation $\sum_{i=1}^{n}(\theta_k^r[i] - \theta_k^*[i]) \cdot \nabla \mathcal{L}_{\text{KD},k}(\theta_k^r)[i]$ is the definition of the inner product between $\theta_k^r - \theta_k^*$ and $\nabla \mathcal{L}_{\text{KD},k}(\theta_k^r)$:

$$\langle\theta_k^r - \theta_k^*, \nabla \mathcal{L}_{\text{KD},k}(\theta_k^r)\rangle = \sum_{i=1}^{n}(\theta_k^r[i] - \theta_k^*[i]) \cdot \nabla \mathcal{L}_{\text{KD},k}(\theta_k^r)[i]. \tag{105}$$

Thus, Term A can be defined as:

$$\mathbb{E}[\langle\theta_k^r - \theta_k^*, \nabla \mathcal{L}_{\text{KD},k}(\theta_k^r, \xi_k^r)\rangle] = \langle\theta_k^r - \theta_k^*, \nabla \mathcal{L}_{\text{KD},k}(\theta_k^r)\rangle. \tag{106}$$

For Term B, we use the assumption 3:

$$\mathbb{E}[\|\nabla \mathcal{L}_{\mathrm{KD},k}(\theta_k^r, \xi_k^r)\|^2] \leq \|\nabla \mathcal{L}_{\mathrm{KD},k}(\theta_k^r)\|^2 + \sigma^2. \tag{107}$$

Therefore, we have:

$$\mathbb{E}[\|\theta_k^{r+1} - \theta_k^*\|^2 \,|\, \theta_k^r] \leq \|\theta_k^r - \theta_k^*\|^2 - 2\eta \langle \theta_k^r - \theta_k^*, \nabla \mathcal{L}_{\mathrm{KD},k}(\theta_k^r) \rangle + \eta^2 \left( \|\nabla \mathcal{L}_{\mathrm{KD},k}(\theta_k^r)\|^2 + \sigma^2 \right). \tag{108}$$

Next, we need to leverage the $\mu$-strong convexity property to establish certain key facts. Since $\mathcal{L}_{\mathrm{KD},k}(.)$ it is $\mu$-strong convexity, by the definition, we have:

$$\mathcal{L}_{\mathrm{KD},k}(\theta') \geq \mathcal{L}_{\mathrm{KD},k}(\theta) + \langle \nabla \mathcal{L}_{\mathrm{KD},k}(\theta), \theta' - \theta \rangle + \frac{\mu}{2} \|\theta' - \theta\|^2. \tag{109}$$

Eq. (109) can be rewritten as:

$$\langle \nabla \mathcal{L}_{\mathrm{KD},k}(\theta_k^r) - \nabla \mathcal{L}_{\mathrm{KD},k}(\theta_k^*), \theta_k^r - \theta_k^* \rangle \geq \mu \|\theta_k^r - \theta_k^*\|^2. \tag{110}$$

The **derivation** of Eq. (110):

Let's define function $\mathcal{L}(\theta)$ is $\mu$-strongly convex if for all $\theta, \theta' \in \mathbb{R}^p$:

$$\mathcal{L}(\theta') \geq \mathcal{L}(\theta) + \langle \nabla \mathcal{L}(\theta), \theta' - \theta \rangle + \frac{\mu}{2} \|\theta' - \theta\|^2. \tag{111}$$

Using the strong convexity inequality at both $\theta$ and $\theta'$, we get two inequalities:

1. At $\theta$:

$$\mathcal{L}(\theta') \geq \mathcal{L}(\theta) + \langle \nabla \mathcal{L}(\theta), \theta' - \theta \rangle + \frac{\mu}{2} \|\theta' - \theta\|^2. \tag{112}$$

2. At $\theta'$:

$$\mathcal{L}(\theta) \geq \mathcal{L}(\theta') + \langle \nabla \mathcal{L}(\theta'), \theta - \theta' \rangle + \frac{\mu}{2} \|\theta - \theta'\|^2. \tag{113}$$

Rearranging these two inequalities, we have:

$$(\mathcal{L}(\theta') - \mathcal{L}(\theta)) \geq \langle \nabla \mathcal{L}(\theta), \theta' - \theta \rangle + \frac{\mu}{2} \|\theta' - \theta\|^2, \tag{114}$$

$$(\mathcal{L}(\theta) - \mathcal{L}(\theta')) \geq \langle \nabla \mathcal{L}(\theta'), \theta - \theta' \rangle + \frac{\mu}{2} \|\theta - \theta'\|^2. \tag{115}$$

Adding these two inequalities together, we have:

$$0 \geq \langle \nabla \mathcal{L}(\theta), \theta' - \theta \rangle + \langle \nabla \mathcal{L}(\theta'), \theta - \theta' \rangle + \mu \|\theta - \theta'\|^2. \tag{116}$$

Simplify:

$$\langle \nabla \mathcal{L}(\theta) - \nabla \mathcal{L}(\theta'), \theta - \theta' \rangle \geq \mu \|\theta - \theta'\|^2. \tag{117}$$

Hence, we have:

$$\langle \nabla \mathcal{L}_{\mathrm{KD},k}(\theta_k^r) - \nabla \mathcal{L}_{\mathrm{KD},k}(\theta_k^*), \theta_k^r - \theta_k^* \rangle \geq \mu \|\theta_k^r - \theta_k^*\|^2. \tag{109}$$

Since at optimum $\theta_k^*$, $\nabla \mathcal{L}_{\mathrm{KD},k}(\theta_k^*) = 0$, we have:

$$\langle \nabla \mathcal{L}_{\mathrm{KD},k}(\theta_k^r), \theta_k^r - \theta_k^* \rangle \geq \mu \|\theta_k^r - \theta_k^*\|^2. \tag{118}$$

Therefore, we have:

$$-2\eta \langle \theta_k^r - \theta_k^*, \nabla \mathcal{L}_{\mathrm{KD},k}(\theta_k^r) \rangle \leq -2\eta\mu \|\theta_k^r - \theta_k^*\|^2. \tag{119}$$

Since function $\mathcal{L}_{\mathrm{KD},k}$ is $\mu$-strong convexity, it also satisfies the **Polyak–Łojasiewicz (PL) condition** with parameter $\mu > 0$ if for all $\theta \in \mathbb{R}^p$. Hence, we have:

$$\frac{1}{2} \|\nabla \mathcal{L}_{\mathrm{KD},k}(\theta)\|^2 \geq \mu(\mathcal{L}_{\mathrm{KD},k}(\theta) - \mathcal{L}_{\mathrm{KD},k}(\theta_k^*)), \tag{120}$$

The **derivation** of Eq. (120):

Applying the definition of $\mu$-strong convexity with $\theta' = \theta_k^*$, we have:

$$\mathcal{L}_{\text{KD},k}(\theta_k^*) \geq \mathcal{L}_{\text{KD},k}(\theta) + \langle \nabla \mathcal{L}_{\text{KD},k}(\theta), \theta_k^* - \theta \rangle + \frac{\mu}{2} \|\theta_k^* - \theta\|^2. \tag{121}$$

Rearranging the above equation, we have:

$$\mathcal{L}_{\text{KD},k}(\theta) - \mathcal{L}_{\text{KD},k}(\theta_k^*) \leq \langle \nabla \mathcal{L}_{\text{KD},k}(\theta), \theta - \theta_k^* \rangle - \frac{\mu}{2} \|\theta - \theta_k^*\|^2. \tag{122}$$

Using the Cauchy-Schwarz inequality, we have:

$$\langle \nabla \mathcal{L}_{\text{KD},k}(\theta), \theta - \theta_k^* \rangle \leq \|\nabla \mathcal{L}_{\text{KD},k}(\theta)\| \|\theta - \theta_k^*\|. \tag{123}$$

Applying Young's inequality $ab \leq \frac{a^2}{2\mu} + \frac{\mu b^2}{2}$ with $a = \|\nabla \mathcal{L}_{\text{KD},k}(\theta)\|$ and $b = \|\theta - \theta_k^*\|$, we have:

$$\|\nabla \mathcal{L}_{\text{KD},k}(\theta)\| \|\theta - \theta_k^*\| \leq \frac{1}{2\mu} \|\nabla \mathcal{L}_{\text{KD},k}(\theta)\|^2 + \frac{\mu}{2} \|\theta - \theta_k^*\|^2. \tag{124}$$

Substituting back, we have:

$$\mathcal{L}_{\text{KD},k}(\theta) - \mathcal{L}_{\text{KD},k}(\theta_k^*) \leq \frac{1}{2\mu} \|\nabla \mathcal{L}_{\text{KD},k}(\theta)\|^2 + \frac{\mu}{2} \|\theta - \theta_k^*\|^2 - \frac{\mu}{2} \|\theta - \theta_k^*\|^2. \tag{125}$$

Simplifying the above equation, we have:

$$\mathcal{L}_{\text{KD},k}(\theta) - \mathcal{L}_{\text{KD},k}(\theta_k^*) \leq \frac{1}{2\mu} \|\nabla \mathcal{L}_{\text{KD},k}(\theta)\|^2. \tag{126}$$

Multiplying both sides by $2\mu$, we have:

$$\mu(\mathcal{L}_{\text{KD},k}(\theta) - \mathcal{L}_{\text{KD},k}(\theta_k^*)) \leq \frac{1}{2} \|\nabla \mathcal{L}_{\text{KD},k}(\theta)\|^2. \tag{127}$$

Therefore, we have:

$$\frac{1}{2} \|\nabla \mathcal{L}_{\text{KD},k}(\theta)\|^2 \geq \mu(\mathcal{L}_{\text{KD},k}(\theta) - \mathcal{L}_{\text{KD},k}(\theta_k^*)). \tag{120}$$

Building on the results above, we proceed to bound $\|\nabla \mathcal{L}_{\text{KD},k}(\theta_k^r)\|^2$ using the $L_s$-smoothness and $\mu$-strong convexity properties of $\mathcal{L}_{\text{KD},k}(.)$.

From the PL condition, we have:

$$\frac{1}{2} \|\nabla \mathcal{L}_{\text{KD},k}(\theta)\|^2 \geq \mu \left( \mathcal{L}_{\text{KD},k}(\theta) - \mathcal{L}_{\text{KD},k}(\theta_k^*) \right). \tag{128}$$

Rearranging the above equation, we have:

$$\|\nabla \mathcal{L}_{\text{KD},k}(\theta)\|^2 \geq 2\mu \left( \mathcal{L}_{\text{KD},k}(\theta) - \mathcal{L}_{\text{KD},k}(\theta_k^*) \right). \tag{119}$$

Since at optimum $\theta_k^*$, $\nabla \mathcal{L}_{\text{KD},k}(\theta_k^*) = 0$, from $L_s$-smoothness, specifically the gradient norm bound, we have:

$$\|\nabla \mathcal{L}_{\text{KD},k}(\theta)\| \leq L_s \|\theta - \theta_k^*\|. \tag{129}$$

Squaring both sides, we have:

$$\|\nabla \mathcal{L}_{\text{KD},k}(\theta)\|^2 \leq L_s^2 \|\theta - \theta_k^*\|^2. \tag{130}$$

From Eq. (120) and Eq. (130), we have:

$$2\mu\left(\mathcal{L}_{\text{KD},k}(\theta) - \mathcal{L}_{\text{KD},k}(\theta_k^*)\right) \leq \|\nabla\mathcal{L}_{\text{KD},k}(\theta)\|^2 \leq L_s^2\|\theta - \theta_k^*\|^2. \tag{131}$$

Hence, we have:

$$2\mu\left(\mathcal{L}_{\text{KD},k}(\theta) - \mathcal{L}_{\text{KD},k}(\theta_k^*)\right) \leq L_s^2\|\theta - \theta_k^*\|^2. \tag{132}$$

Dividing both sides by $2\mu$, we have:

$$\mathcal{L}_{\text{KD},k}(\theta) - \mathcal{L}_{\text{KD},k}(\theta_k^*) \leq \frac{L_s^2}{2\mu}\|\theta - \theta_k^*\|^2. \tag{133}$$

Hence, we have:

$$\|\nabla\mathcal{L}_{\text{KD},k}(\theta)\|^2 \leq 2L_s\left(\mathcal{L}_{\text{KD},k}(\theta) - \mathcal{L}_{\text{KD},k}(\theta_k^*)\right) \leq 2L_s\cdot\left[\frac{L_s^2}{2\mu}\|\theta - \theta_k^*\|^2\right] = \frac{L_s^3}{\mu}\|\theta - \theta_k^*\|^2. \tag{134}$$

For simplicity, define $C = \dfrac{L_s^3}{\mu}$, so

$$\|\nabla\mathcal{L}_{\text{KD},k}(\theta)\|^2 \leq C\|\theta - \theta_k^*\|^2. \tag{135}$$

Next, we use the above results to simplify the Eq. (108), we get:

$$\mathbb{E}[\|\theta_k^{r+1} - \theta_k^*\|^2\mid\theta_k^r] \leq \|\theta_k^r - \theta_k^*\|^2 - 2\eta\underbrace{\langle\theta_k^r - \theta_k^*,\ \nabla\mathcal{L}_{\text{KD},k}(\theta_k^r)\rangle}_{\text{gradient alignment}} + \eta^2\left(\|\nabla\mathcal{L}_{\text{KD},k}(\theta_k^r)\|^2 + \sigma^2\right)$$

$$\leq \|\theta_k^r - \theta_k^*\|^2 - 2\eta\mu\|\theta_k^r - \theta_k^*\|^2 + \eta^2 C\|\theta_k^r - \theta_k^*\|^2 + \eta^2\sigma^2 \tag{136}$$

$$= \left(1 - 2\eta\mu + C\eta^2\right)\|\theta_k^r - \theta_k^*\|^2 + \eta^2\sigma^2.$$

The second inequality used the fact that $-2\eta\langle\theta_k^r - \theta_k^*, \nabla\mathcal{L}\rangle \leq -2\eta\mu\|\theta_k^r - \theta_k^*\|^2$ (Eq. (119)) and the bound $\|\nabla\mathcal{L}_{\text{KD},k}(\theta_k^r)\|^2 \leq C\|\theta_k^r - \theta_k^*\|^2$ from Eq. (135).

To simplify the above equation, we define:

$$\gamma = (1 - 2\eta\mu + C\eta^2), \quad \beta = \eta^2\sigma^2. \tag{137}$$

Finally, we expand Eq. (136) to derive its global (unconditional) expectation version. Specifically, we define the global (unconditional) expectation of the squared distance at iteration $r$ as:

$$\mathcal{E}^r := \mathbb{E}[\|\theta_k^r - \theta_k^*\|^2], \tag{138}$$

From Eq. (136), we have:

$$\mathbb{E}\left[\|\theta_k^{r+1} - \theta_k^*\|^2\mid\theta_k^r\right] \leq \gamma\|\theta_k^r - \theta_k^*\|^2 + \beta. \tag{139}$$

Since, we have: $\mathbb{E}\left[\mathbb{E}[X\mid Y]\right] = \mathbb{E}[X]$, by taking expectation on both sides, we have:

$$\mathbb{E}\left[\|\theta_k^{r+1} - \theta_k^*\|^2\right] \leq \gamma\mathbb{E}\left[\|\theta_k^r - \theta_k^*\|^2\right] + \beta, \tag{140}$$

Hence, we can get:

$$\mathcal{E}^{r+1} = \mathbb{E}[\|\theta_k^{r+1} - \theta_k^*\|^2] \leq \gamma\mathcal{E}^r + \beta. \tag{141}$$

We now write Eq. (141) for each $r = 0, 1, \dots$.

In particular:

$$\begin{aligned}
\mathcal{E}^1 &\leq \gamma\mathcal{E}^0 + \beta, \\
\mathcal{E}^2 &\leq \gamma\mathcal{E}^1 + \beta \leq \gamma\left(\gamma\mathcal{E}^0 + \beta\right) + \beta = \gamma^2\mathcal{E}^0 + \gamma\beta + \beta = \gamma^2\mathcal{E}^0 + (\gamma + 1)\beta, \\
\mathcal{E}^3 &\leq \gamma\mathcal{E}^2 + \beta \leq \gamma\left(\gamma^2\mathcal{E}^0 + (\gamma + 1)\beta\right) + \beta = \gamma^3\mathcal{E}^0 + (\gamma^2 + \gamma)\beta + \beta = \gamma^3\mathcal{E}^0 + (\gamma^2 + \gamma + 1)\beta.
\end{aligned} \tag{142}$$

Observing the pattern, we find:

$$\mathcal{E}^r \;\leq\; \gamma^r\,\mathcal{E}^0 \;+\; (\gamma^{r-1} + \gamma^{r-2} + \cdots + \gamma + 1)\,\beta. \tag{143}$$

Therefore, we have

$$\mathcal{E}^r = \mathbb{E}[\|\theta_k^r - \theta_k^*\|^2] \;\leq\; \gamma^r\,\|\theta_k^0 - \theta_k^*\|^2 + \sum_{\tau=0}^{r-1} \gamma^\tau\,\beta \tag{144}$$

This completes the proof of Theorem 2. $\qquad\square$

## C. Additional Experiment Materials

### C.1. Additional Experiments on the Hurricane Dataset at $\psi = 0.7$

To demonstrate the performance improvement achieved by satellites other than those three evaluated in the main text, we conduct six additional experiments on the Hurricane dataset at $\psi = 0.7$. Table 4 reports the test accuracies (%) for these six additional satellites. The results confirm that both FOL-A and FOL maintain consistent performance gains across a broader range of clients.

Table 4. Test accuracies (%) for six additional clients on the *Hurricane* dataset with $\psi = 0.7$.

| Dataset | Hurricane | | | | | |
|---|---|---|---|---|---|---|
| Satellite # | 41 | 3 | 9 | 22 | 56 | 51 |
| Methods | | | $\psi = 0.7$ | | | |
| Local | 90.45 | 82.35 | 88.63 | 90.67 | 86.01 | 91.18 |
| FOL-A (E=1) | 94.27 | 91.18 | 92.73 | 93.10 | 93.87 | 96.57 |
| FOL-A (E=2) | 95.54 | 94.12 | 93.64 | 93.68 | 95.16 | 97.06 |
| FOL-A (E=3) | 96.18 | 96.06 | 94.09 | 95.40 | 95.74 | 97.55 |
| FOL (E=1) | 93.11 | 85.29 | 90.02 | 91.95 | 89.81 | 93.63 |
| FOL (E=2) | 93.63 | 91.33 | 91.82 | 92.53 | 90.07 | 94.12 |
| FOL (E=3) | 94.27 | 93.04 | 92.27 | 94.25 | 91.92 | 95.59 |
| DENSE | 70.02 | 67.35 | 68.13 | 71.31 | 69.57 | 70.16 |
| Co-Boosting | 74.61 | 69.16 | 72.51 | 73.63 | 75.21 | 74.47 |

To demonstrate convergence of clients under our proposed FOL method, we provide the loss-vs-epoch curves for the six additional satellites during distillation in Figure 1. All six clients exhibit a rapid initial decrease in training loss and steadily converge to a stable low-loss plateau, confirming the robustness of the FOL distillation process across diverse satellites.

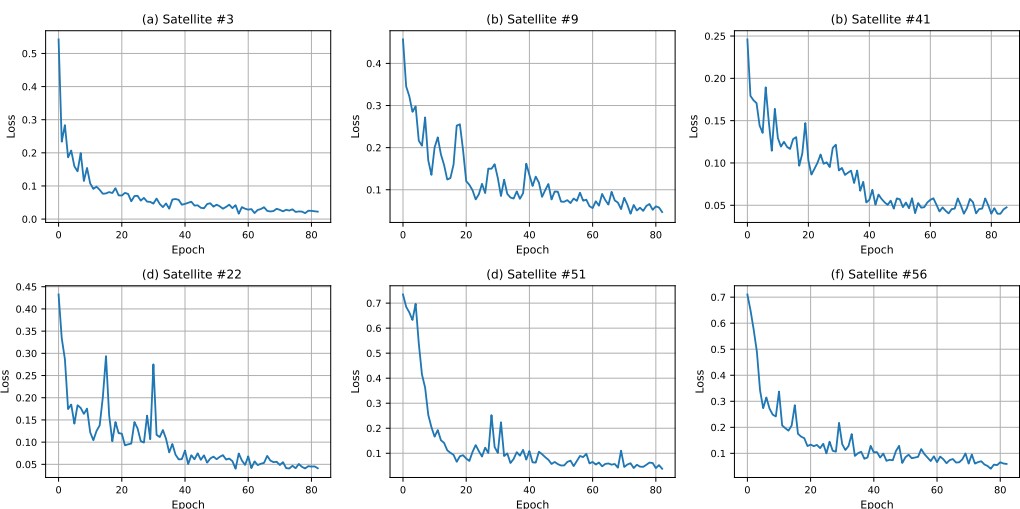

Figure 1. Training loss convergence over 83 epochs for six additional Hurricane clients at $\psi = 0.7$.

## C.2. Additional Hyperparameter Settings

In our experiments on CIFAR-10, CIFAR-100, SVHN, and the satellite datasets (Wildfire and Hurricane), we set $\lambda_p = 0.1$, $\gamma_{\text{shared}} = 0.05$, $\gamma_{\text{unshared}} = 0.02$ in Equation (5), and we set the distillation regularization weight in Equation (12) to $0.01$. These values were selected via cross-validation and were found to consistently yield robust performance, effectively balancing alignment, diversity retention, and model personalization.

## C.3. Summary of Datasets

Note that, by default, some datasets provide separate training, validation, and testing subsets. To streamline the presentation and ensure consistency, we merged the validation samples into the testing dataset in the table below.

*Table 5.* Summary of datasets used in the experiments.

| Name | #Training Samples | #Testing Samples | #Classes | Image Size |
|------|-------------------|------------------|----------|------------|
| Wildfire | 30,250 | 12,600 | 2 | 350×350×3 |
| Hurricane | 10,000 | 4,000 | 2 | 128×128×3 |
| CIFAR-10 | 50,000 | 10,000 | 10 | 32×32×3 |
| CIFAR-100 | 50,000 | 10,000 | 100 | 32×32×3 |
| SVHN | 73,257 | 26,032 | 10 | 32×32×3 |

## C.4. Performance of DENSE and Co-Boosting on the Wildfire and Hurricane Datasets

The original authors of DENSE and Co-Boosting have extensively evaluated their methods on various image classification benchmark datasets, consistently demonstrating that these methods achieve optimal performance when $n = 5$. In this study, we adopt their methods, evaluation metrics (testing the model on the entire testing dataset), and default configurations to further assess their performance. Specifically, we evaluate whether these methods retain their optimal performance on real-world satellite image datasets when $n = 5$. To simulate non-IID data distributions, the datasets were divided among 5 and 10 clients using a Dirichlet distribution with parameter $\psi = 0.7$ for the binary classification tasks Wildfire and Hurricane. For each configuration, results are reported as the average of 5 runs with different random seeds.

*Table 6.* Test Accuracy (%) of the Final Server Model on the Wildfire and Hurricane Datasets.

| Dataset | Wildfire | | Hurricane | |
|---------|----------|--|-----------|--|
| Method | $n = 5, \psi = 0.7$ | $n = 10, \psi = 0.7$ | $n = 5, \psi = 0.7$ | $n = 10, \psi = 0.7$ |
| DENSE | 89.87 ± 1.32 | 83.38 ± 1.57 | 69.35 ± 1.69 | 63.95 ± 1.55 |
| Co-Boosting | 92.35 ± 0.86 | 85.46 ± 1.12 | 75.45 ± 1.27 | 68.05 ± 1.42 |

## C.5. Neural Network Structures for Each Dataset

*Table 7.* Cifar-10 Network Structure (Model Architecture for Each Base Model).

| Layer (type) | Input Shape | Output Shape | Param # |
|--------------|-------------|--------------|---------|
| Conv2d | [1, 3, 224, 224] | [1, 128, 222, 222] | 3,584 |
| MaxPool2d | [1, 128, 222, 222] | [1, 128, 111, 111] | 0 |
| Conv2d | [1, 128, 111, 111] | [1, 128, 109, 109] | 147,584 |
| MaxPool2d | [1, 128, 109, 109] | [1, 128, 54, 54] | 0 |
| Conv2d | [1, 128, 54, 54] | [1, 128, 52, 52] | 147,584 |
| Linear | [169, 2048] | [169, 10] | 20,490 |

*Table 8.* Network structure for Hurricane and CIFAR-100 Datasets.

| Dataset | Structure | Usage |
|---------|-----------|-------|
| CIFAR-100 | ResNet18 (Non-pretrained) | Model architecture for each base model |
| Hurricane | ResNet18 (Non-pretrained) | Model architecture for each base model |

*Table 9.* Wildfire network structure (Model Architecture for Each Base Model).

| Layer (type) | Input Shape | Output Shape | Param # |
|---|---|---|---|
| Conv2d | [1, 3, 224, 224] | [1, 64, 224, 224] | 1,792 |
| BatchNorm2d | [1, 64, 224, 224] | [1, 64, 224, 224] | 128 |
| ReLU | [1, 64, 224, 224] | [1, 64, 224, 224] | 0 |
| MaxPool2d | [1, 64, 224, 224] | [1, 64, 112, 112] | 0 |
| Conv2d | [1, 64, 112, 112] | [1, 128, 112, 112] | 73,856 |
| BatchNorm2d | [1, 128, 112, 112] | [1, 128, 112, 112] | 256 |
| ReLU | [1, 128, 112, 112] | [1, 128, 112, 112] | 0 |
| MaxPool2d | [1, 128, 112, 112] | [1, 128, 56, 56] | 0 |
| Conv2d | [1, 128, 56, 56] | [1, 256, 56, 56] | 295,168 |
| BatchNorm2d | [1, 256, 56, 56] | [1, 256, 56, 56] | 512 |
| ReLU | [1, 256, 56, 56] | [1, 256, 56, 56] | 0 |
| MaxPool2d | [1, 256, 56, 56] | [1, 256, 28, 28] | 0 |
| Conv2d | [1, 256, 28, 28] | [1, 512, 28, 28] | 1,180,160 |
| BatchNorm2d | [1, 512, 28, 28] | [1, 512, 28, 28] | 1,024 |
| ReLU | [1, 512, 28, 28] | [1, 512, 28, 28] | 0 |
| MaxPool2d | [1, 512, 28, 28] | [1, 512, 14, 14] | 0 |
| AdaptiveAvgPool2d | [1, 512, 14, 14] | [1, 512, 1, 1] | 0 |
| Flatten | [1, 512, 1, 1] | [1, 512] | 0 |
| Linear | [1, 512] | [1, 2] | 1,026 |

## C.6. Selected Satellites and Dataset Sizes

The table below summarizes the selected satellites and their corresponding dataset sizes, including the number of training, validation, and testing samples used for evaluation under $\psi = 0.7$. For the Wildfire dataset, satellites #13, #28, and #48 were chosen, with training dataset sizes ranging from 693 to 2,627 samples. Similarly, for the Hurricane dataset, satellites #35, #32, and #44 were selected, with training dataset sizes ranging from 732 to 1,507 samples.

*Table 10.* Selected Satellites and Dataset Sizes for Evaluation.

| | $\psi = 0.7$ | | |
|---|---|---|---|
| **Dataset Name** | **Satellite #** | **#Training** | **#Validation** | **#Testing** |
|---|---|---|---|---|
| | 13 | 1,488 | 319 | 320 |
| Wildfire | 28 | 2,627 | 563 | 564 |
| | 48 | 693 | 148 | 149 |
| | 35 | 1,507 | 323 | 323 |
| Hurricane | 32 | 732 | 157 | 158 |
| | 44 | 778 | 167 | 167 |

