# OpenReview forum: "Federated Oriented Learning: A Practical One-Shot Personalized Federated Learning Framework"
_ICML.cc/2025/Conference — ICML 2025 poster_

### Official Review · Reviewer_bQPg · 2025-03-11

**Overall Recommendation:** 2

**Summary:**

The paper introduces Federated Oriented Learning (FOL), a novel one-shot personalized federated learning (OPFL) framework designed for communication-constrained environments such as LEO satellite networks. FOL integrates multi-stage processes—fine-tuning, structured pruning with alignment regularization, ensemble refinement, and knowledge distillation—to enable clients to adaptively integrate knowledge from neighboring models under stringent communication constraints. Theoretical guarantees on empirical risk discrepancy and convergence are provided. Extensive experiments on wildfire, hurricane, CIFAR-10, and CIFAR-100 datasets demonstrate FOL’s superiority over baselines like FedAvg, DENSE, and Co-Boosting, achieving accuracy improvements.

**Claims And Evidence:**

The assumption that the local validation set has the same distribution as the final test set, and using it to determine the contribution of each model in the ensemble, can lead to overfitting on the final test set. This is unrealistic in practical scenarios.

**Essential References Not Discussed:**

I think some related decentralized learning works should be cited,

**Experimental Designs Or Analyses:**

1. FOL involves multiple stages (e.g., fine-tuning, pruning), but the paper does not compare training time or communication costs against baselines. For resource-constrained environments, this omission limits practical applicability assessment.

2.  In the experiments of this paper, the authors emphasize the performance of the ensemble model, but the ultimate goal of the algorithm should be a single personalized model. Furthermore, the performance of naive local training is actually quite similar to the personalized model produced by the algorithm, especially considering the significant increase in storage and computational costs introduced by the proposed method. If the 30 times the storage space used for the ensemble model were instead allocated to improving the local model, the performance of the local model could potentially be further enhanced.

**Methods And Evaluation Criteria:**

The communication cost of peer-to-peer communication between all nodes and their neighboring nodes needs to be experimentally compared with the communication cost in a federated learning setup with a central server.

**Other Comments Or Suggestions:**

In the problem statement part, “Given an image classification task” may be not necessary. As far as I know, other PFL articles usually clarify the specific task in the experiment instead of problem setting. Personally, I think deleting this sentence makes the method look more generalized rather than being limited in a specific domain. Of course, this is just a suggestion, not a necessity.

**Other Strengths And Weaknesses:**

The authors focus on a setting where there is no central server, and clients communicate directly with their neighbors, which represents a decentralized learning framework. This field has already been extensively studied, and the paper should reference related works in this area.

In the proposed setup, where clients directly communicate model parameters with their neighbors, serious privacy concerns arise. Additionally, the assumption that the local validation set has the same distribution as the final test set, and using it to determine the contribution of each model in the ensemble, can lead to overfitting on the final test set. This is unrealistic in practical scenarios.

The definition of "neighbor models" in the paper lacks sufficient detail. Specifically, it remains unclear how the adjacency or relationship between neighbors is determined and whether it changes over time.

The communication cost of peer-to-peer communication between all nodes and their neighboring nodes needs to be experimentally compared with the communication cost in a federated learning setup with a central server.

In the experiments of this paper, the authors emphasize the performance of the ensemble model, but the ultimate goal of the algorithm should be a single personalized model. Furthermore, the performance of naive local training is actually quite similar to the personalized model produced by the algorithm, especially considering the significant increase in storage and computational costs introduced by the proposed method. If the 30 times the storage space used for the ensemble model were instead allocated to improving the local model, the performance of the local model could potentially be further enhanced.

**Questions For Authors:**

The details have been discussed in the strengths and weaknesses part.

**Relation To Broader Scientific Literature:**

The authors focus on a setting where there is no central server, and clients communicate directly with their neighbors, which represents a decentralized learning framework. This field has already been extensively studied, and the paper should reference related works in this area.

**Theoretical Claims:**

No.I haven't checked the details in the proofs. Broadly speaking, such assumptions and results are reasonable.

---

> ### Author Rebuttal · Authors · 2025-03-28
>
> **1. Concern About the Local Validation Set Has the Same Distribution as the Final Test Set.**
>
> **Response:** We would like to clarify that, in personalized learning, it is standard practice to assume that the local validation set and the test set follow the same distribution. In real-world scenarios, such as LEO satellite constellations, each client’s data is collected by the client itself, and the training, validation, and test sets are just uniformly sampled from these data. This naturally implies that the distribution of these validation and test sets are aligned.
>
>
> **2. Definition of "Neighbor Models" and Communication Cost Comparison.**
>
> **Response:** Regarding the definition of neighbor models, we clarify that in our framework a neighbor is any client reachable via a one-hop connection. For example, in a Starlink LEO satellite network, each satellite continuously broadcasts beacons, and any satellite that receives a beacon is considered a neighbor.
>
> Regarding the communication costs, we respectfully clarify that in our one-shot setting, each client exchanges models only once with its one-hop neighbors, incurring significantly lower communication overhead than the iterative rounds required by FL setup with a central server.
>
> **3. Personalized Model Performance and Ensemble Storage Use.**
> *Furthermore, the performance of naive local training  is actually quite similar to......, If the 30 times the storage space used....*
>
> **Response:** The reviewer seems to have mis-read the performance comparison between the naive local model and our personalized model shown in Tables 1-3 in the paper. We respectfully clarify that our final personalized model consistently outperforms naive local training. As shown in these tables, FOL improves accuracy by up to 13.95\% on Wildfire ($\psi$ = 0.5), 30.16\% on Hurricane ($\psi$ = 0.3), 6.77\% on CIFAR-10 ($\psi$ = 0.7), and 9.01\% on CIFAR-100 ($\psi$ = 0.7). These consistent and substantial gains confirm that our method effectively extracts and integrates valuable knowledge from neighbor models.
>
> As for the suggestion of using the storage budget (30 times of model size) to train a larger local model instead of using them to fuse neighbors' models, we note that increasing model size alone can potentially increase the risk of overfitting and cannot achieve the benefits of aggregating (new) knowledge from non-IID clients. Additionally, we also want to point out that the 30-times storage cost is only needed at the ensemble step. After the ensemble model is distilled, the final personalized model has the same size as a standard local model.
>
> **4.Privacy Concerns.**
>
>  **Response:** In this work, our primary focus is on improving model accuracy, and thus the issue of privacy is out of the scope of this paper. In real-world applications such as Starlink LEO satellite networks, those satellites typically belong to the same operator, therefore there is no privacy issue. Moreover, if privacy is required in other applications, homomorphic encryption can be integrated into the parameter sharing process, enabling computations directly on encrypted data without compromising performance.
>
> **5. Training Time or Communication Costs Against Baselines.** *The communication cost of peer-to-peer communication......needs to be experimentally compared with....*
>
> **Response:** We want to point out that our proposed FOL method has lower computation and communication overhead than DENSE and Co-Boosting (i.e., the baselines). Specifically,   DENSE and Co-Boosting rely on a modified GAN structure in which the generator is composed of multiple large fully connected layers and runs for 30 epochs per distillation epoch over a total of 200 epochs (i.e, in total 6000 epochs for GAN alone). In FOL, the computation cost of various components are as follows:  fine tuning: 30 epochs, structured pruning: 30 epochs, post fine tuning: 30 epochs, ensemble refinement: 10 epochs, knowledge distillation: at most 500 epochs. In total: at most 600 epochs, which is much smaller than that of DENSE and Co-Boosting.  Moreover, these additional processes incur only local computational cost, with no extra communication overhead.
>
>
> **6. Related Work In Decentralized Learning.**
>
> **Response:** In the revised version, we will add additional citations and discussion of relevant decentralized learning approaches. It is important to note that to the best of our knowledge, no existing decentralized learning work has provided a one-shot personalized features. Our work is the first to provide such a feature with both theoretical guarantees and experiment validation.

---

### Official Review · Reviewer_hUmY · 2025-03-13

**Overall Recommendation:** 2

**Summary:**

•	In order to address the situation of limited client communication in federated learning, this paper introduces a novel federated learning paradigm - OPFL and presents a four-stage one-shot PFL algorithm FOL (Federated Oriented Learning). FOL can learn a personalized model for each client without the need of central server to generate a global model. The convergence analysis is also discussed in the paper. Under the scenario of Low Earth Orbit (LEO), FOL can demonstrate good performance on the Wildfire and Hurricane satellite image datasets, as well as on CIFAR10 and CIFAR100.

**Claims And Evidence:**

In the reality of Low Earth Orbit, each client can often only communicate with neighboring clients, and adding a global server for all Low Earth Orbits can be very expensive. This paper attempts to address this situation.

**Essential References Not Discussed:**

The authors' research on relevant papers is relatively comprehensive, and they give a relatively complete introduction to one-shot Federated Learning and Personalized Federated Learning, which are most relevant to this paper.

**Experimental Designs Or Analyses:**

The authors mainly focus on the validation of satellite datasets. The SVHN dataset is used in both DENSE and Co-Boosting papers, but this paper does not use this dataset. I think this paper should validation the method on more datasets.

**Methods And Evaluation Criteria:**

For the communication situation faced by Low Earth Orbit (LEO), FOL can learn a good model for each orbit relatively well.

**Other Comments Or Suggestions:**

There are some writing errors in the paper, such as using the same hyperparameter λ in equations (5) and (12), but the hyperparameters are different.
In P5 ’The proof of Theorem 2 is provided in Appendix B.2.’. Perhaps it should be 'The proof of Theorem 1 is provided in Appendix B.3.'

**Other Strengths And Weaknesses:**

Weakness:
(1) This paper should consider the fairness issue in federated learning.
This paper assumes a setting where each client can only communicate with its neighboring clients. Consider the following scenario: there are three groups of clients, K1, K2, and K3, which cannot communicate internally among themselves. Each client in K1 and K3 can only communicate once, while K1 can communicate with K2, K2 can communicate with K3, but K1 cannot communicate directly with K3.
If all clients in K1 first communicate with all clients in K2, and then all clients in K2 communicate with all clients in K3, when the number of clients in K1 and K3 is the same, each client in these groups will have the same communication cost. However, each client in K1 can only collect knowledge from clients in K2, while clients in K3 can collect knowledge from both K1 and K2 (since K2 processes and aggregates K1's knowledge). Therefore, at the same communication cost, K3 obtains more knowledge than K1. This disparity becomes even greater when there are more clients in K1. It is unfair to them.
(2) In P4, 'Each gating parameter \alpha_{l,i}\in [0,1] controls the retention or pruning of the i-th filter or neuron.'Aaccording to this, \alpha_{l,i}) is the gating parameter, which is a real number in [0,1], such as 0.5. This is equivalent to scaling each weight to a certain extent, and this is not a common pruning process. Common pruning methods either set the weights to 0 or retain the weights.

**Questions For Authors:**

In equation (5), there are hyperparameters \lambda, \gamma_shared, \gamma_unshared. How to set the value for the hyperparameters in equation (5) and equation (12) is not mentioned in the paper. I did not see any experimental results on them.

**Relation To Broader Scientific Literature:**

This paper may be helpful for federated learning in satellites. The authors mainly conduct experiments on relevant satellite datasets. Prior to this, there have been some works on one-shot federated learning, but they main focus on learning a global model. This paper proposes to learn a personalized model for each client in one-shot federated learning.

**Theoretical Claims:**

I mainly checked the correctness of Theorem 2, and the main part of the theoretical proof in the paper is correct, with some minor writing issues.
(1) Equation (28) has not been fully expressed.
(2) In equation (17), the parameter L is defined as the parameter of L-smoothness, but in the Proof of Theorem 1 of B.3, the parameter for the L-Lipschitz condition is also L.

---

> ### Author Rebuttal · Authors · 2025-03-28
>
> **1. Fairness Issue in the One-Shot Communication Setting.**
>
> **Response:** Please note that for personalized learning, fairness does not mean equal accuracy across individual users, but instead it means every user has comparable opportunity to improve its accuracy (i.e., opportunity of learning). Under this definition of fairness, the major issue in the learning example given by the reviewer, i.e., $(t_0: K1 ⟷  K2), (t_1: K2 ⟷  K3)$, where $t_1>t_0$ and $(t: a ⟷ b)$ denotes a learning at time $t$ between users $a$ and $b$ (because they meet at that moment), is that it only considers a short interval $[t_0, t_1]$ in the learning period of these users while has artificially neglected what could happen after that interval, e.g., $K_1$ may later meet and learn from someone who has learned from $K_3$ earlier. Specifically, consider the following example sequence of learning that extends the reviewer's example: $(t_0: K1 ⟷  K2), (t_1: K2 ⟷  K3), (t_2: K3⟷  K4), (t_3: K1⟷  K4)$. So, at moment $t_3$, $K1$ makes up its knowledge on $K3$ by learning from $K4$ who just learned from $K3$ at moment $t_2$. To make our above example more concrete, we have conducted experiment on the aforementioned sequence of learning, and list the accuracy of $K1$ at different moments in Table 1. From this table, it can be observed that the accuracy of $K1$ was improved at $t_0$ and $t_3$, indicating that $K1$ indeed obtained the opportunity of improving its accuracy by learning at these two moments, comparable to the learning opportunities possessed by $K3$. In general, by considering the more realistic scenario where users encounter with each other over a wider time horizon, the fairness issue raised by the reviewer will diminish due to the knowledge propagation among users.
>
> **Table: Accuracy of $K1$ at different moments over Hurricane and CIFAR-10.**
>
> | Methods                                     | Hurricane ($\psi=0.3$) | CIFAR-10 ($\psi=0.7$) |
> |---------------------------------------------|-------------------------|------------------------|
> | Initial (i.e., Local)                       | 82.14\%                 | 60.47\%                |
> | FOL-A ($t_0: K1 ⟷  K2$)      | 91.07\%                 | 63.18\%                |
> | FOL-A ($t_3: K1 ⟷  K4$)      | 92.86\%                 | 67.15\%                |
> | FOL ($t_0: K1 ⟷  K2$)        | 85.71\%                 | 61.91\%                |
> | FOL ($t_3: K1 ⟷  K4$)        | 89.07\%                 | 63.13\%                |
>
> **2. Gating Parameters and Pruning Mechanism.**
>
> **Response:** Compared with the common **hard** pruning method, wherein weights are either strictly set to zero or fully retained based on a fixed/unified threshold,  our **soft** pruning has the unique advantage of adaptive, fine-grained selection of threshold to prune for each individual connections (i.e., threshold $\alpha_l$ for weight $W_l$). These gating parameters/thresholds are optimized during training, allowing the model to selectively prune less important connections, which ultimately results in a more robust model. This strategy is also commonly used in recent differentiable pruning methods.
>
> **3.Hyperparameter Settings in Equations (5) and (12).**
>
> **Response:** In our experiments, for CIFAR-10, CIFAR-100, and the satellite datasets (Wildfire and Hurricane), we set $\lambda = 0.1$, $\gamma_{\text{shared}} = 0.05$, and $\gamma_{\text{unshared}} = 0.02$ in Equation (5), and we set the distillation regularization weight in Equation (12) to $0.01$. These values were selected via cross-validation and were found to consistently yield robust performance, effectively balancing alignment, diversity retention, and model personalization. We will include these information  in the final paper.
>
> **4. Minor Writing Issues.**
>
> **Response:** We acknowledge that Equation (28) was not fully expressed, that the same constant “L” is used for both the L-smoothness and L-Lipschitz conditions, and that the same notation $\lambda$ appears in Equations (5) and (12). In the revised manuscript, we will update the notation to clearly distinguish between the different constants and ensure that all equations are fully expressed. Importantly, these issues do not affect the correctness of the proofs, as confirmed by our analysis and by the reviewer’s own inspection of Theorem 2.
>
> **5. Suggestion to Include SVHN.**
>
> **Response:** We want to point out that our validation in the paper is based on more comprehensive and domain-relevant datasets than those done for DENSE and Co-Boosting. Specifically, none of those datasets used to validate DENSE and Co-Boosting in their original papers are relevant to satellite applications, which are the typical applications our proposed FOL method targets for. In contrast, the wildfire and hurricane datasets used in our paper are more relevant to the satellite application of the proposed FOL method.

---

### Official Review · Reviewer_DHoW · 2025-03-14

**Overall Recommendation:** 3

**Summary:**

This paper first introduces an important limitation of existing Personalized Federated Learning methods, which is the need of multiple communication rounds to update models. This will lead to massive communication costs and impracticable for the real-world scenarios. Moreover, the authors argue that personalizing the global model is not feasible in practice, as the global model produced by federated learning (FL) algorithms typically lacks the adaptive modules necessary for effective local adaptation. Based on this, the authors propose a novel algorithm for one-shot personalized federated learning (PFL), called Federated Oriented Learning (FOL), which operates in a decentralized manner and enables clients to iteratively enhance their local models by learning from their neighbors through a single round of local model communication. FOL consists of several key stages: model pretraining, model collection, fine-tuning, pruning, post fine-tuning, ensemble refinement, and knowledge distillation. Additionally, the authors establish two theoretical guarantees: one on the empirical risk discrepancy between the student and teacher models, and another on the convergence of the distillation process.

The paper demonstrates a well-motivated and innovative approach; however, certain critical steps are described in a vague and potentially misleading manner, which could hinder the clarity of the methodology. It is recommended that the authors enhance the readability of these sections to ensure a more precise and accessible presentation. Additionally, the complexity of the proposed method appears to be notably high, raising concerns about its practical applicability. A more detailed discussion on computational efficiency and scalability would greatly strengthen the paper.

**Claims And Evidence:**

1.	The authors provide a thorough and detailed analysis of existing research and the problem at hand in the introduction section, effectively highlighting the significance of the issue and presenting a well-justified motivation.

**Essential References Not Discussed:**

NA

**Experimental Designs Or Analyses:**

1.	The authors claim that their method is effective for clients with highly diverse datasets; however, the experiments do not include any scenarios that validate this claim.

2.	Given that the Ensemble Model  (FOL-A) consistently outperforms other approaches in the experimental results, including full FOL, it raises the question of whether the Knowledge Distillation component should be removed. Alternatively, is there a more effective personalization strategy that could replace knowledge distillation?

3.	The number of baseline methods appears to be somewhat limited.

**Methods And Evaluation Criteria:**

1.	Each client collects the local models of its neighbors and performs operations such as fine-tuning and models alignment by structured pruning on its own dataset. Essentially, every step involves training all the neighbor’s local models on the local dataset, which is undoubtedly unsuitable for resource-constrained devices. Moreover, the authors do not clarify whether each local model is trained on the local dataset only once or for multiple epochs. The authors should reduce unnecessary computational burdens to enhance the practical applicability of the proposed method.

2.	Optimal Weighted Ensemble described in Section 3.3 appears to be misleading and requires further clarification. In Line 231, it seems to suggest assembling models to form a new model, but later descriptions indicate that this step is more akin to ensembling logits. Additionally, the statement that **"the ensemble outcome (a.k.a. the teacher model) is K times larger"** is unclear, as Equation (10) implies that the size of the outcome remains unchanged, with the corresponding elements being weighted sums.

**Other Comments Or Suggestions:**

See above.

**Other Strengths And Weaknesses:**

See above.

**Questions For Authors:**

See above.

**Relation To Broader Scientific Literature:**

NA

**Theoretical Claims:**

1.	The paper would benefit significantly from a convergence analysis of Equation (5), as it would strengthen the theoretical foundation of the proposed method.

---

> ### Author Rebuttal · Authors · 2025-03-28
>
> **1. Optimal Weighted Ensemble Clarification.** *Optimal Weighted Ensemble described in Section 3.3 appears to be misleading...; Additionally, the statement that "the ensemble outcome (a.k.a. the teacher model) is K times larger...*
>
> **Response:** We respectfully clarify that the "Optimal Weighted Ensemble" in Section 3.3 is performed entirely at the logit level. In our approach, we aggregate the outputs (logits) of the K selected base models using a weighted sum (as shown in Equation (10)), so the ensemble’s final output retains the same dimensionality as that of a single model. More formally, let vector $\mathbf{g}_i = [g_i^{(1)}, \ldots, g_i^{(N)}]$
>
>  denote the $N$-dimensional logit output of base model $i$, then the output of the ensemble model can be written as $\mathbf{G} = \sum_{i=0}^K w_i \mathbf{g}_i$, where $K$ is the number of base models participating in the ensemble.
>
> Furthermore, in our original text, the phrase “K times larger” referred to the fact that the ensemble model is composed of K base models, and hence the number of parameters in the ensemble model is roughly $K$ times of the size of a base model.  It was never intended to imply that the final output dimension (i.e., the dimensionality of $\mathbf{G}$) increases by a factor of $K$.
>
> **2. Role of Knowledge Distillation.** *Given that the Ensemble Model (FOL-A) consistently outperforms ... it raises the question ... should be removed. Alternatively, is there a more effective personalization strategy that could replace knowledge distillation?*
>
> **Response:** The knowledge distillation component is necessary in order to ensure the final personalized model has the same size as the initial local model. In particular, the ensemble step FOL-A bloats the model size by $K$ time, the distillation step compresses the bloated model by a factor of $K$, leading to a final model roughly at the same size of the initial model.
>
> Furthermore, based on our best knowledge, knowledge distillation achieves the best performance among all other personalization strategies in one‑shot federated learning scenarios.
>
> **3. Computational Complexity and Practical Applicability.** *Each client collects the local models of its neighbors and performs operations...; Moreover, the authors do not clarify whether each local model is trained on the local dataset only once or for multiple epochs...*
>
> **Response:** We want to point out that our proposed FOL method has much lower computation and communication overhead than DENSE and Co-Boosting (i.e., the baselines). Specifically, DENSE and Co-Boosting rely on a modified GAN structure in which the generator is composed of multiple large fully connected layers and runs for 30 epochs per distillation epoch over a total of 200 epochs (i.e, in total 6000 epochs for GAN alone). In FOL, the computation cost of various components are as follows:  fine tuning: 30 epochs, structured pruning: 30 epochs, post fine tuning: 30 epochs, ensemble refinement: 10 epochs, knowledge distillation: at most 500 epochs. In total: at most 600 epochs, which is much smaller than that of DENSE and Co-Boosting.  Moreover, these additional processes incur only local computational cost, with no extra communication overhead. Furthermore, we note that FOL is designed for one-shot communication scenarios, e.g., LEO satellite networks or intermittent IoT systems, for which limited communication bandwidth, instead of the computation power, is the primary constraint.
>
>
> **4. Experiments on Data Diversity and Baseline Selection.** *The authors claim that their method is effective for clients with highly diverse datasets...; The number of baseline methods appears to be somewhat limited.*
>
> **Response:** In our experimental settings, we use a Dirichlet distribution to partition datasets across 70 clients, thereby explicitly simulating highly non-IID conditions with varying degrees of data heterogeneity (with $\psi$ values ranging from 0.1 to 0.7).
>
> Regarding baseline methods, we compare against widely recognized state‑of‑the‑art one‑shot federated learning approaches, namely DENSE and Co‑Boosting. We believe that our experimental design thoroughly demonstrates the effectiveness of our method.
>
>
>
>
> **5.Suggestion to Include the Convergence Analysis of Equation (5).**
>
> **Response:** We want to point out that structured pruning is just one of the several intermediate steps within FOL. Instead of proving the convergence of this intermediate step, we have proved the convergence of the final step of the proposed FOL method in Theorem 2 of the paper. We believe proving the convergence of the final step is more meaningful and important than just proving the convergence of an intermediate step. In our extensive experiments, we have observed that all intermediate steps consistently converge.
>
> **6. Minor Writing Issues Eq. (28).**
>
> **Response:** We appreciate the reviewer pointing out this typo and will ensure all equations are correctly presented in the final paper.

---

> > ### Comment · Reviewer_DHoW · 2025-04-04
> >
> > Thank you for the rebuttal. After carefully reading the reviews from others and the corresponding rebuttals, some of my concerns have been addressed. However, I still believe there is a lot of room for improvement. Therefore, I  decide to keep my score.

---

> > > ### Author Response · Authors · 2025-04-06
> > >
> > > Dear Reviewer DHoW,
> > >
> > > Thank you so much for providing your comments to our rebuttal! We are glad to see that we have addressed some of your concerns. If you have any additional concern that is related to the generalizability of the proposed FOL model, we just want to inform you that we have conducted new experiments to validate the proposed model and compared it to the state of the art DENSE, Co-Boosting, and FedAvg on a new dataset SVHN, for which the results are shown in the following Table. It can be observed that FOL-A consistently achieves the highest accuracy, and FOL (E=3) surpasses the best baseline (in this case Co-Boosting) by up to 9.24\% over the SVHN dataset. Now the proposed FOL model has been validated and compared with DENSE and Co-Boosting over 5 datasets: Hurricane, Wildfire, CIFAR-10, CIFAR-100, and the newly added SVHN. We hope these new experiment results have better demonstrated the generalizability of the proposed FOL model, and therefore has adequately addressed any concern related to the model's generalizability. If you have any additional concerns or suggestions, we will be more than happy to further address/accommodate them. Thank you!
> > >
> > > **Table: Test accuracies (%) on SVHN, $\psi = 0.5$, reported as mean ± std.**
> > > **Dataset:** SVHN
> > > **Satellite #:** 21
> > >
> > > | Method           | Accuracy (%)       |
> > > |------------------|--------------------|
> > > | Local            | 78.97 ± 1.75       |
> > > | FOL-A (E=1)      | 85.73 ± 1.63       |
> > > | FOL-A (E=2)      | 86.26 ± 1.28       |
> > > | FOL-A (E=3)      | **88.37 ± 0.92**   |
> > > | FOL (E=1)        | 81.09 ± 1.54       |
> > > | FOL (E=2)        | 81.62 ± 1.36       |
> > > | FOL (E=3)        | 82.85 ± 1.18       |
> > > | FOL-AN (E=1)     | 79.62 ± 1.64       |
> > > | FOL-AN (E=2)     | 80.92 ± 1.41       |
> > > | FOL-AN (E=3)     | 83.15 ± 1.33       |
> > > | FOL-N (E=1)      | 80.04 ± 1.74       |
> > > | FOL-N (E=2)      | 80.39 ± 1.39       |
> > > | FOL-N (E=3)      | 81.15 ± 1.22       |
> > > | DENSE            | 69.53 ± 1.57       |
> > > | Co-Boosting      | 73.58 ± 1.48       |
> > > | FedAvg (E=1)     | 53.08 ± 2.32       |
> > > | FedAvg (E=2)     | 58.72 ± 1.79       |
> > > | FedAvg (E=3)     | 55.49 ± 1.93       |

---

### Official Review · Reviewer_rxR9 · 2025-03-25

**Overall Recommendation:** 5

**Summary:**

This paper proposes Federated Oriented Learning, a novel framework for One-Shot Personalized Federated Learning designed for environments with constrained or infrequent communication or limited contact windows. The authors further provide two theoretical guarantees on empirical risk discrepancy between student and teacher models and the convergence of the distillation process.

**Claims And Evidence:**

Most claims are supported, with real-world and benchmark datasets and theoretical contribution

**Essential References Not Discussed:**

Some multi-round methods could be mentioned here.

**Experimental Designs Or Analyses:**

The experimental settings are valid for OPFL with class imbalance and natural non-IID splits settings.
Ablation studies demonstrates the contribution of each component.
Considers heterogeneity and real-world dataset.

I wonder if evaluation is on 1–3 clients per setting, which might seems insufficient for personalization-focused FL research where client-wise variance matters.

**Methods And Evaluation Criteria:**

The multi-stage pipeline is methodologically valid and appropriate for OPFL. Non-IID data is considered with different levels of heterogeneity.

**Other Comments Or Suggestions:**

N/A

**Other Strengths And Weaknesses:**

Strength:
Paper is well-written and experiments carefully designed with theoretical analysis.

Weaknesses:
A figure showing the trend of convergence is desired here.

**Questions For Authors:**

How does FOL scale with increased client count or larger models?

**Relation To Broader Scientific Literature:**

This paper is closed connected to FL, PFL and OFL, with model distillation and pruning.

Integrating one-shot FL, pruning with alignment, and distillation for personalization seems novel.

**Theoretical Claims:**

Theoretical contribution includes (1) risk discrepancy between student and teacher model with KL-divergence and (2) convergence of distillation with standard theoretical assumptions.

---

> ### Author Rebuttal · Authors · 2025-03-28
>
> **1. Regarding Evaluation on 1–3 Clients per Setting.**
>
> **Response:** The accuracy of 6 additional clients are given in following Table. Similar performance trends as those 3 shown in our original paper can be observed in this table (i.e., those 3 shown in our original table indeed are representative). Together with these 6 additional clients, we have shown the performance of 9 clients. We hope this has addressed the concern of the reviewer.
>
>
> **Table:** Test accuracies (%) for six additional clients on the *Hurricane* dataset with $\psi$ = 0.7.
>
> | **Dataset**     |               |          | **Hurricane** |          |               |          |
> |-----------------|---------------|----------|---------------|----------|---------------|----------|
> | **Satellite #** | **41**        | **3**    | **9**         | **22**   | **56**        | **51**   |
> | **Methods**     |               |          | **$\psi = 0.7$** |          |               |          |
> | Local           | 90.45%        | 82.35%   | 88.63%        | 90.67%   | 86.01%        | 91.18%   |
> | FOL-A (E=1)     | 94.27%        | 91.18%   | 92.73%        | 93.10%   | 93.87%        | 96.57%   |
> | FOL-A (E=2)     | 95.54%        | 94.12%   | 93.64%        | 93.68%   | 95.16%        | 97.06%   |
> | FOL-A (E=3)     | 96.18%        | 96.06%   | 94.09%        | 95.40%   | 95.74%        | 97.55%   |
> | FOL (E=1)       | 93.11%        | 85.29%   | 90.02%        | 91.95%   | 89.81%        | 93.63%   |
> | FOL (E=2)       | 93.63%        | 91.33%   | 91.82%        | 92.53%   | 90.07%        | 94.12%   |
> | FOL (E=3)       | 94.27%        | 93.04%   | 92.27%        | 94.25%   | 91.92%        | 95.59%   |
> | DENSE           | 70.02%        | 67.35%   | 68.13%        | 71.31%   | 69.57%        | 70.16%   |
> | Co-Boosting     | 74.61%        | 69.16%   | 72.51%        | 73.63%   | 75.21%        | 74.47%   |
>
> **2. Suggestion to Include a Convergence Trend Figure.**
>
> **Response:** Because the rebuttal policy of ICML does not allow us to upload figures as part of our rebuttal, we present a representative trace of the loss value in the training of the distilled FOL model in the following table. The convergence trend of the training can be clearly observed in the table. We will present the same trace as a figure in our final paper, as suggested by the reviewer.
>
>
> **Table:** FOL distillation training loss vs. epoch id for one of the satellites over the Hurricane dataset.
>
> | **Epoch**         | 1      | 2      | 3      | 4      | 5      | 6      | 7      | 8      | 9      |
> |-------------------|--------|--------|--------|--------|--------|--------|--------|--------|--------|
> | **Training Loss** | 0.135  | 0.1001 | 0.0818 | 0.1109 | 0.097  | 0.0968 | 0.0839 | 0.0859 | 0.0891 |
>
>
> **3. Concerning the Scalability of FOL.**
>
> **Response:** Our proposed FOL framework is designed to scale efficiently. The storage, computation, and communication cost do not change with the number of clients. These costs increase proportionally with the size of local model.

---

### Decision · Program_Chairs · 2025-05-01

**Decision:**

Accept (poster)

**Comment:**

The paper proposes FOL, to iteratively enhance the local training via utilizing neighboring models. The authors also propose theoretical analysis regarding the discrepancy and the distillation.